# Ultra-narrowband and rainbow-free mid-infrared thermal emitters enabled by a flat band design in distorted photonic lattices

Kaili Sun [1], Yangjian Cai [1], Lujun Huang [2] ✉ & Zhanghua Han [1] ✉

Most reported thermal emitters to date employing photonic nanostructures to achieve narrow bandwidth feature the rainbow effect due to the steep dispersion of the involved high-Q resonances. In this work, we propose to realize thermal emissions with high temporal coherence but free from rainbow effect, by harnessing a novel flat band design within a large range of wavevectors. This feature is achieved by introducing geometric perturbations into a square lattice of high-index disks to double the period along one direction. As a result of the first Brillouin zone halving, the guided modes will be folded to the Γ point and interact with originally existing guided-mode resonances to form a flat band of dispersion with overall high Q. Despite the use of evaporated amorphous materials, we experimentally demonstrate a thermal emission with the linewidth of 23 nm at 5.144 μm within a wide range of output angles (from −17.5° to 17.5°).

All objects above the absolute temperature of zero Kelvin emit energy in the form of thermal radiations. According to Stefan-Boltzmann and Planck's radiation law, the intensity of thermal radiation is proportional to the temperature of the object and its spectral distribution will shift towards higher frequencies as the temperature increases. The naturally generated thermal radiation is typically broadband, omnidirectional, incoherent and unpolarized. In recent years, considerable attention has been given to tailoring thermal emissions. The introduction of various artificial micro/nanostructures allows for flexible control of thermal emission properties (such as polarization, coherence, etc)[1–4]. By carefully designing these structures to control the near-field interactions, the ability to manipulate far-field electromagnetic radiations can be greatly enhanced. This has led to a variety of important applications, such as infrared sensing[5,6], near-field thermal management[7,8], radiative cooling[9], thermo-photovoltaics[10], imaging[11], and infrared camouflage[12,13]. In particular, thermal emission with steered properties has recently garnered significant attention because they can offer both fundamental interest, e.g., higher temporal and spatial coherence, and energy-more-efficient solutions to many practical applications. For example, traditional non-dispersive

infrared (NDIR) system[14] often use microlight bulbs as light sources to detect the absorption of various gases and compounds. However, many target substances have sharp absorption spectrum, and require the incorporation of narrowband filters in the NDIR system, which not only increases the system complexity, but also greatly reduces energy utilization efficiency since much of the radiations is filtered out and wasted. Therefore, it is crucial to develop narrowband (which suggests high temporal coherence) thermal emitters as energy-more-efficient substitutes with improved spectral selectivity.

Currently, various micro/nanostructures and physical mechanisms have been employed to achieve narrowband thermal emitters. In a ground-breaking study in 2002, Greffet et al. demonstrated that surface phonon polaritons (SPhPs) supported by the polar material of silicon carbide (SiC) can help achieve coherent thermal emission[15]. Later, Inampudi et al. conducted a detailed study on the unidirectional emission performance of thermal emitters based on this platform[16,17]. However, these emitters can only be designed for a limited operating range (e.g., 10.5–12.5 μm for SiC) referred to as the Reststrahlen band[18,19]. Another approach involves using the moiré effect supported by a dual-layer twisted SiC grating to achieve narrowband tunable

[1]Shandong Provincial Key Laboratory of Optics and Photonic Devices, Center of Light Manipulation and Applications, School of Physics and Electronics, Shandong Normal University, Jinan 250358, China. [2]School of Physics and Electronic Science, East China Normal University, Shanghai 200241, China. ✉e-mail: ljhuang@phy.ecnu.edu.cn; zhan@sdnu.edu.cn

thermal emission[20]. However, the experimental realization of this complex geometric structure poses significant challenges. The bull's eye grating made from metals, as a standalone structure for realizing a narrowband and directional thermal emission by manipulating the propagation of surface plasmon polaritons (SPPs), has also been studied in recent years[21]. Unfortunately, the relatively large propagation loss of surface waves including both SPhPs and SPPs sets an upper bound for the emission temporal coherence. To date, the most widely used structure in this direction is the metal-insulator-metal (MIM) sandwich geometry[22,23], thanks to its fabrication simplicity and non-dispersion characteristics. However, due to the high inherent loss of metals, these emitters have a broad spectral emission linewidth ($Q < 20$). In recent years, it appears as an emerging and promising topic to achieve ultra-narrow band thermal emitters[24] by coupling the broadband thermal fluctuations in heated metals to the high-Q resonances[25] supported by all-dielectric nanostructures. Since only the spectral component matching the resonance can be extracted and re-directed to free space, thermal emissions with the bandwidth as low as a few tens of nanometers can be realized even using evaporated amorphous dielectric materials[26,27]. Two typical high-Q resonances in all-dielectric nonlocal metasurfaces are the bound states in the continuum (BICs)[28,29] and the quasi-guided modes (QGM)[30]. BICs, initially proposed by Von Neumann and Wigner in quantum mechanics in 1929, occupy few discrete optical states in the continuum region of the energy-momentum space. The decoupling of these states from free space results in an infinite radiation Q-factor ($Q_{rad}$) and vanishing resonance linewidth. In general, both intrinsic (by breaking the structural symmetry) and extrinsic (by changing the excitation angle) perturbations can transform BICs into quasi-BICs (QBICs)[31], enabling the realization of ultra-narrow resonance linewidths for practical applications. QGMs originate from folding of the dispersion diagram of broadband infinite-Q guided modes (GMs) to the continuum when the period-increasing perturbation is introduced into the photonic lattice[27,32]. As an exceptional type of guided-mode resonance (GMR), the QGMs not only exhibit perturbation-dependent Q-factors but also advantageously feature high robustness of the Q-factor against the frequency/wavenumber over the QBICs[33]. Unfortunately, both the QBICs and QGMs achieved in photonic lattices are still optical states embedded in some steep dispersion bands. For example, the BICs in photonic crystal slabs can be considered as eigenstates due to the coupling between different modes[34]. The QGMs are derivatives of the guided modes (GMs) and thus inherit similar yet folded dispersion band of the predecessor GMs. So, the broadband and omnidirectional thermal fluctuations in the heated metals may couple into all those modes, reaching thermal emissions with the wavelength highly sensitive to the output angle, i.e., the rainbow effect. If one aspires for a high spatial coherence of the thermal emission, a steep dispersion is desired. In that case, one frequency only corresponds to few spatial/Fourier components, which will work in phase to offer a long spatial coherence length[32]. One can even manipulate the FBZ folding to make the thermal emitter operating at a steeper part of the dispersion band in order to achieve a higher spatial coherence. However, in many practical applications, one cares more about the bandwidth or in other words temporal coherence. For thermal emitters with steep dispersion, due to the rainbow effect, it becomes necessary to use some

spatial filters to select the spectral component of interest, which not only increases the system complexity but also implies a waste of radiation energy. In addition, since thermal radiations are usually weak, large samples are preferred to increase the output power, and one normally uses lenses to collect the emitted signals within a large collection angle to fully make use of all the radiations from the whole structure. Due to the rainbow effect, multiple resonances will be collected and the overlap of these spectral components will inevitably lead to a broader resonance, significantly deteriorating the temporal coherence. In this context, a flat band design to host many high-Q optical resonances all operating at the same frequency but a large wide of wavenumbers will help address the problem, and the development of this kind of rainbow-free thermal emitters becomes necessary.

The flat band design has been extensively studied in fundamental optical sciences due to the large compatibility of all the spatial components at the same frequency under wide-angle illumination, as well as the associated slow light effect which can greatly increase the interaction time between light and matter. However, a simultaneous realization of flat band dispersion and high Q-factor can be challenging. Table 1 presents the compared performance of flat band designs in periodic photonic structures achieved by different approaches. The use of moiré structures to achieve flat bands is conceptually appealing[35–37], with some reported experimental confirmation of low-threshold micro-lasers[38]. However, the complex fabrication process and uncontrolled radiation characteristics make it difficult to be widely adopted in practical applications. Nguyen et al. attempted to manipulate energy-momentum dispersion relationships through symmetry breaking[39,40], but the large breaking makes it difficult to maintain the high-Q property. Plasmonic BICs have been demonstrated to achieve angle-insensitive flat band effects[41,42], but the intrinsic losses of metals make it challenging to achieve high Q-factors. Some researchers have used high-order Mie resonances in all-dielectric metasurfaces to achieve wavefront shaping[43], thereby obtaining high numerical aperture lenses. However, the Q-factor obtained by Mie resonance is also relatively low. Therefore, it still remains an unsolved challenge to achieve both high Q-factors and flat band effects within a wide range of wavenumbers.

In this work, we introduce an effective approach that, through a simple structure, enables the realization of the flat band around the Γ point. By combining this effect with a controllable high-Q QBIC mode, we have achieved an ultra-narrow band and output-angle-insensitive mid-infrared (MIR) thermal emitter in both numerical calculation and experimental demonstrations. The key in the design is the introduction of a perturbation into the square lattice of a single-disk structure (SDS) to trigger a doubling of the period along one direction. In that case, the SDS will evolve to a double-disk structure (DDS) with two disks within one unit cell. This transformation will lead to the folding of dispersion bands originally located at the edges of the FBZ (GMs) to be close to the Γ point in the distorted lattice. The strong coupling between the newly-generated QGMs with the originally existing GMRs along both $k_x$ and $k_y$ directions will lead to significant band splitting. By adjusting the coupling strength, a flat band with low group velocity can be obtained within a wide range of wavevectors. By further introducing some asymmetry between two disks in the DDS unit cell, the QBIC resonance can be generated, forming stably high Q-factors for

**Table 1 | Comparison of performance of flat dispersion bands achieved by different methods**

| Structures | Fabrication accuracy requirements | Q-factor | Q or spectral tunability | Ref. |
|---|---|---|---|---|
| Moiré photonic crystal | High | High | Low | 35 |
| Symmetry breaking grating | Low | Low | High | 39 |
| Plasmonic BICs | Low | Low | High | 41 |
| Mie-resonant metasurfaces | Low | Low | High | 43 |
| Distorted photonic lattices | Low | High | High | This work |

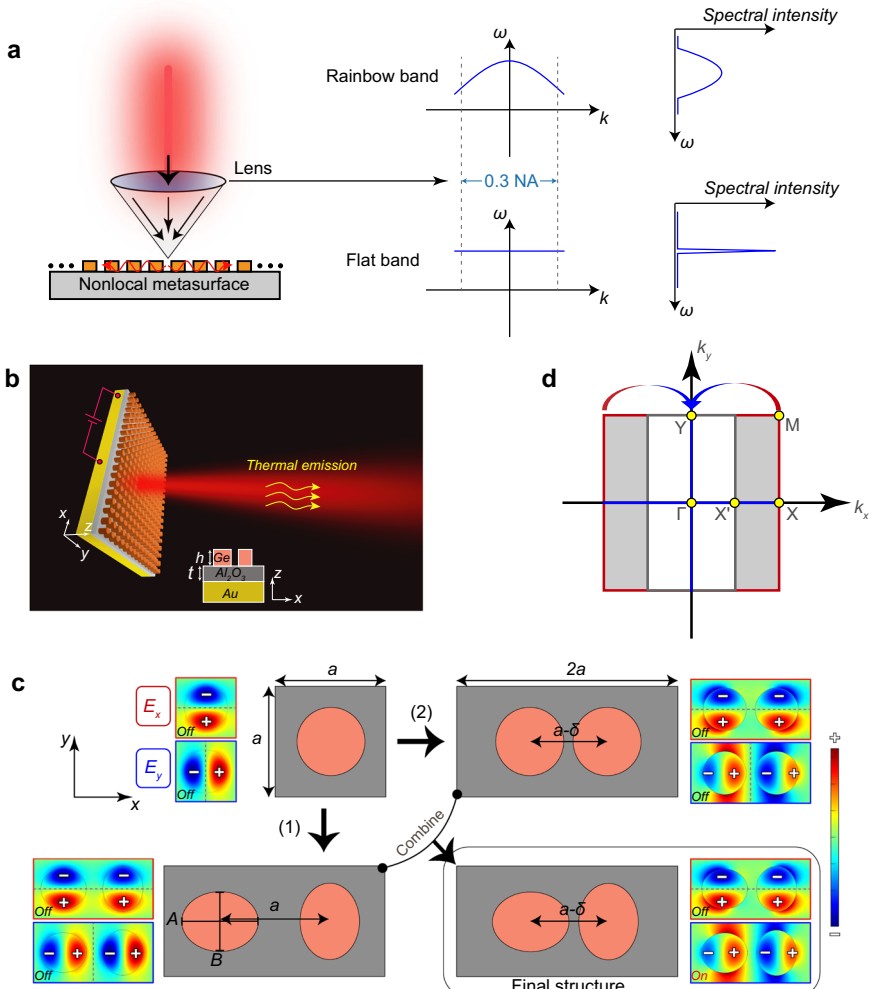

**Fig. 1 | Flat band design in distorted photonic lattices. a** Schematic diagram illustrates the spectral response of high-Q nonlocal metasurfaces with different dispersion behaviors when excited by a focused beam. **b** Artistic rendering of the thermal emitter, where the red output beam represents the thermal radiation at the QBIC resonance. **c** Illustrates the evolution process from the traditional SDS to the final DDS structure, with red and blue block diagrams representing the real parts of $E_x$ and $E_y$ and the dashed line representing the axis of symmetry in the polarization direction. **d** The corresponding shrinking of the FBZ due to the introduction of the perturbation.

wavevectors close to the Γ point. Leveraging this design gives us a thermal emitter operating at a wide angle within the ultra-narrowband wavelength range. The structure was then manufactured using standard nanofabrication techniques with subsequent measurement of the emission properties, confirming its superior performance of ultra-narrow bandwidth at the same central wavelength over a broad range of output angles.

## Results

Figure 1a briefly illustrates the distinctive spectral responses of high-Q nonlocal metasurfaces that support different dispersion behaviors. For those high-Q modes within steep dispersion bands, the resonance frequency has a strong dependence on the wavevector. When the metasurface of this kind is illuminated by a focused light beam, the excitation of multiple resonances will overlap and give rise to a broadband response in the far-field spectra. For an ideal flat band, however, the modes share almost the same frequency within a wide range of wavevectors, which helps maintain the narrow-band characteristics in the far-field spectrum. The slow-light effect enabled by this flat dispersion can also enhance the interaction between light and matter, which facilitates the coupling of thermal fluctuations into the nonlocal metasurfaces, thus providing an

important foundation for achieving ultra-narrowband and rainbow-free thermal emitters.

## Numerical results

Figure 1b illustrates artistically the geometry of the designed thermal emitter, where the bottommost yellow region represents a sufficiently thick gold as the conductive layer, while the top layer uses a high refractive index and low-loss *Ge* metasurface to provide the nonlocal response of high-Q resonances. The $Al_2O_3$ serves as the intermediate low-refractive index buffer layer, and it has two main roles. On one hand, it isolates the nonlocal metasurface from the metal to reduce mode losses resulting from the dissipation in gold. On the other hand, its thickness can be adjusted to control the impedance match between the emitter and the environment, thus optimizing the emissivity. All the material parameters used in the numerical calculations can be found in the Supplementary Information (Part I). When an electric current is applied to the bottom metal to heat the sample or the device is accommodated by a high-temperature environment, randomly oriented dipoles associated with thermal excitations in the metal will couple to the high-Q modes supported by the upper structure, thus achieving steered thermal emission. The numerical simulations were performed based on the finite element method (FEM) using the

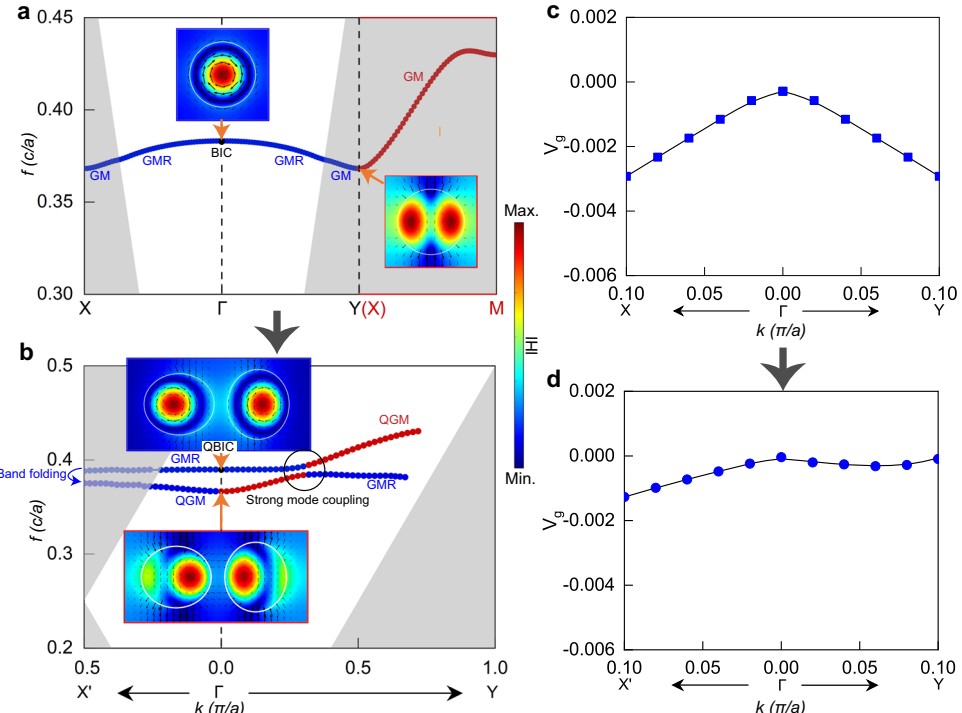

**Fig. 2 | Dispersion analysis of the distorted photonic lattices. a** Shows the dispersion band supported by SDS along different directions within the FBZ, respectively, and (**b**) presents the band in the new FBZ after the transformation from SDS to DDS. The insets show the amplitude distribution of the magnetic field and the vector distribution of the electric field. The black circle represents the wavevector region of strong mode coupling. The white areas in (**a**) and (**b**) represent the continuum. Panels **c** and **d** are the corresponding group velocities $v_g$ close to the Γ point for modes supported by SDS and DDS, respectively.

commercial software of COMSOL Multiphysics. Floquet periodic boundary conditions were implemented in the lateral directions of a unit cell to define the wavevectors, while perfect matching layers (PML) were employed in the $z$ direction to mimic a nonreflecting infinite domain. Based on Kirchhoff's law of thermal radiation, the properties of emissivity and absorptivity of a thermally balanced object are identical. Hence, the emissivity can be characterized by the calculated absorptivity of the structure, i.e., $E(\lambda) = A(\lambda) = 1 - R(\lambda) - T(\lambda)$. Due to the optically thick bottom layer of gold to prevent light transmission ($T(\lambda) = 0$), it can be further simplified as $E(\lambda) = A(\lambda) = 1 - R(\lambda)$. The evolution process of the structural design shown in Fig. 1c is used to demonstrate the underlying physics. We use the following geometric parameters for the square lattice of disk array as an example: $a = 2\,\mu m$, $h = 0.6\,\mu m$, $t = 0.45\,\mu m$, the circular disk diameter $D = 1.45\,\mu m$, which results in the resonance frequencies located in the MIR range. The calculated dispersion band of the supported modes in this lattice is presented in Fig. 2a. At small wavevectors close to the Γ point, GMR bands are supported along both ΓX and ΓY directions. However, each GMR band extends across the light line at larger wavenumbers close to the X and Y points, where the GMRs switches to GMs. Particularly, the dispersion band along the XM direction continuing from the X point is all below the light line (see the red line in Fig. 2a), indicating that it is a complete GM band. Due to the $C_4$ symmetry of the lattice, the ΓX and ΓY directions have the same band structures. In addition, the GMR and GM bands have the same frequency at the two high-symmetry points of X and Y. The distributions of the electromagnetic field at the Γ point for the GMR and at the X point for the GM are shown in the inset of Fig. 2a, revealing the properties of magnetic dipoles along the $z$-axis.

We further use the symmetry of the electric field components, $E_x$ and $E_y$, to describe the radiation characteristics of the structure. For the disk array with $C_4$ symmetry in the original square lattice shown in the upper left corner of Fig. 1c, the real parts of both components exhibit a perfect anti-symmetric distribution along the symmetric axis

(dashed line) in the corresponding polarization direction, making it a topological singularity[44] (See Fig. S3) at the Γ point. This is a prominent characteristic of symmetry-protected BIC (SP-BIC) which cannot radiate into free space. For the next step, it is required to introduce a period-doubling perturbation into the square lattice to manipulate the FBZ along the $k_x$ direction. There are two ways of triggering this perturbation. One is to change the circular disks to elliptical ones and rotate every second column of them by 90° without changing the center-to-center distance 'a', as shown by operation (1) in Fig. 1c. The other, as shown by operation (2), is to change the center-to-center distance to 'a−δ' between adjacent disks without changing the disk geometry. It is worth noting that although both methods can reach the period doubling in the $x$-direction, the evolved DDS still retains its $C_2$ symmetry, either around the center between two disks or around the center of each disk. In both cases, the real parts of both electric field components still exhibit a perfectly anti-symmetric distribution along the structural symmetric axis (dashed line) in their respective polarization direction. Thus, these modes cannot couple to external plane waves which have even distributions of the electric field. For the DDS implemented by either process (1) or (2), the far-field polarization vector cannot be defined and still manifests as a vortex center carrying +1 topological charge, as shown in Fig. S3, suggesting the nature of BIC with infinite $Q_{rad}$ factors. To break the $C_2$ symmetry of the structure, we combine these two operations by introducing elliptical shapes with different orientations and changing the center-to-center distance 'a−δ' at the same time. The final design of the structure is shown in the bottom right panel of Fig. 1c. In this case, although the $E_x$ field distribution still maintains a perfect anti-symmetry with respect to the $x$-axis, the structure is no longer symmetrical along the $y$ direction, and the $E_y$ mode distribution is disrupted, allowing for coupling with $y$-polarized plane waves. This can be confirmed by its far-field polarization (see Fig. S3d), indicating the conversion of BIC into QBIC. It exhibits a nonlocal resonant mode with a high yet finite Q-factor.

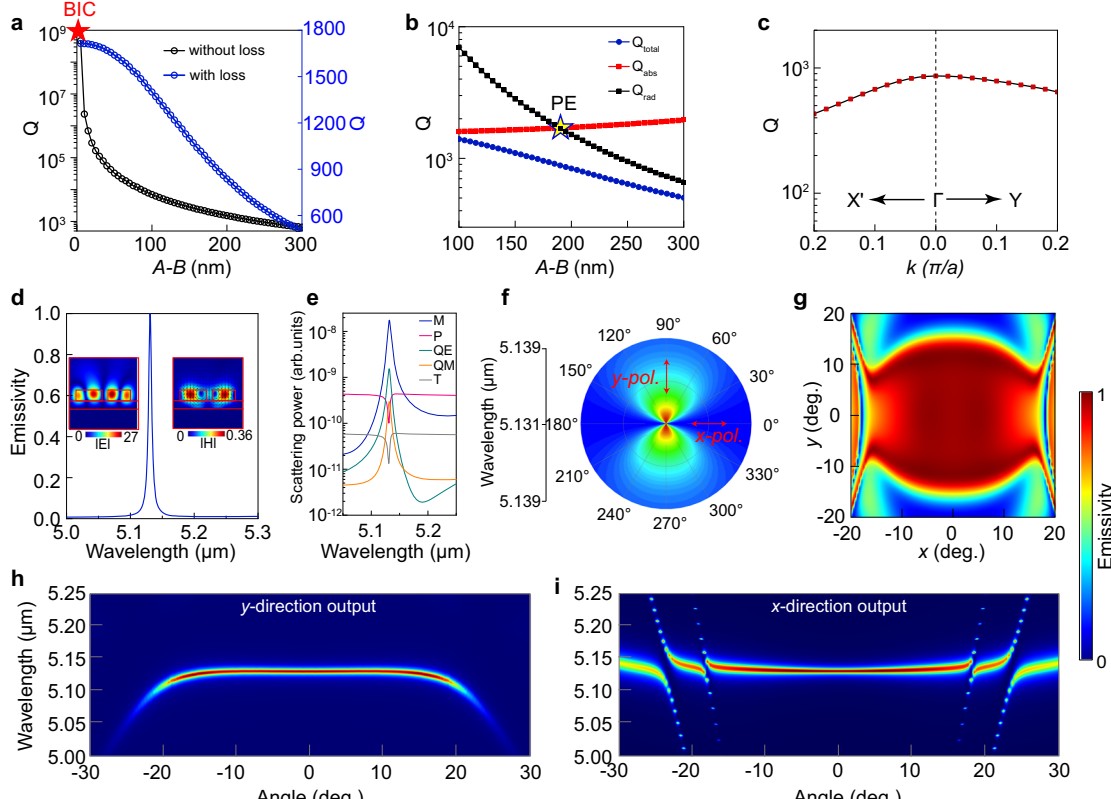

**Fig. 3 | Numerical analysis of the thermal emitter performances. a** Dependence of the Q-factor on the difference $A−B$ ($\Delta$) between the long and short axes of the elliptical disks. The blue curve corresponds to the results with evaporated amorphous materials, while the black curve is for the results where the losses of all materials are neglected. **b** The relationship between $Q_{total}$ (blue), $Q_{abs}$(red), and $Q_{rad}$(black) at different values of $\Delta$. $Q_{rad}$ decreases with increasing $\Delta$. Perfect emission is achieved when $Q_{abs} = Q_{rad}$, as shown by the yellow star symbol, which corresponds to $\Delta = 190$ nm used in the final thermal emitter. **c** The calculated Q-factor at the perfect emission wavelength as a function of wavevectors along the

$\Gamma$Y and $\Gamma$X' directions. **d, e** Represent the emission spectrum and multipole decomposition of the resonance in the normal output direction, respectively, indicating the ultra-narrow linewidth and the main contribution from the magnetic dipole. The insets in (**d**) illustrate the distribution and amplitude of the electric and magnetic fields at the peak wavelength. **f** The emissivity at the emission peak in (**d**) along different directions is used to indicate the y-polarized property. **g** The emissivity at the normal output wavelength (5.131 μm) as a function of output angle in 3D space. **h, i** Represent a 2D mapping of the emissivity as a function of the output angle and wavelength in both the $y$ and $x$ directions, respectively.

When period-doubling perturbations are introduced in the $x$-direction, a distorted photonic lattice is formed, as shown in Fig. 1d, where the FBZ shrinks into a white rectangular region. This transformation folds the dispersion band of the GM along the XM direction in the original FBZ to the $\Gamma$Y direction (i.e., QGM now)[45] in the new lattice, as shown by the red band in the $\Gamma$Y direction in Fig. S4a. Furthermore, the GM part at larger wavenumbers along the $\Gamma$X direction will also be folded back to the $\Gamma$ point, as shown in the folded band in Fig. 2b. Therefore, there will be two dispersion bands in both the $k_x$ and $k_y$ directions in the new FBZ, one is QGM and the other is the originally existing GMR. The strong coupling between the two bands along each direction gives rise to the splitting of them into two new bands. A clear avoided-crossing behavior is seen in Fig. 2b along the $\Gamma$Y direction, which is a signature of the strong-coupling effect. The field distributions at the $\Gamma$ point for the two new bands are presented in the inset of Fig. 2b, which clearly shows that they retain the modes of the original BIC and GM. Importantly, the coupling strength and the slopes of the two new bands can be adjusted by changing the geometry like the center-to-center distance '$a−\delta$' or height $h$ of the two disks (see Fig. S4). The flat-band behavior can then be achieved in one of the two new bands. Through careful design, we have adopted the following perturbation parameters: $\delta = 0.25$ μm, elliptical disk long axis $A = 1.54$ μm and short axis $B = 1.35$ μm. The new band in the upper part of Fig. 2b shows a pronounced flat band behavior in a wide wavevector range. We further solved for the group velocity using $v_g = df/dk$ close to the $\Gamma$ point for the GMR band both in the SDS and the flat band in the DDS,

and the results are shown in Fig. 2c, d, respectively. Clearly, for the GMR band in the SDS, the group velocity rapidly increases at larger wavevectors, reaching $0.003(c/\pi)$ at a wavevector of $0.1\pi/a$. In contrast, for the DDS, $v_g$ remains almost constant and close to 0. In the latter section, we show that such an ultra-low group velocity plays a crucial role in achieving narrowband thermal emissions free from the rainbow effect.

Based on the above theoretical framework, we conducted a detailed study on the emission characteristics of the DDS in Fig. 2b. First, we evaluate the dependence of the Q-factor on the asymmetry of the elliptical disks characterized by the difference between the long/short axes, $\Delta = A−B$, in the absence of material losses. For emissions in the normal direction, without considering the absorption loss of all the materials, the Q-factor exhibits an infinitely high value at $\Delta = 0$, validating the nature of the BIC. As $\Delta$ increases, it shows an inversely quadratic decaying relationship. In our experimental demonstration, to simplify the fabrication, all the films were obtained by using electron beam evaporation (EBE). Compared to traditional single-crystal materials, the obtained films were in amorphous states and thus incurred non-negligible dissipation (see Part I of the Supplementary Information). However, even considering the absorption loss of all materials, the Q-factor can still reach the order of $10^3$ (see Fig. 3a). In practical case, the total Q-factor ($Q_{total}$) of a thermal emission has two contributions from $Q_{rad}$ and $Q_{abs}$ by $1/Q_{total} = 1/Q_{rad} + 1/Q_{abs}$, where $Q_{rad}$ and $Q_{abs}$ characterize the energy dissipation rates caused by the radiation and absorption losses, respectively[46]. According to the

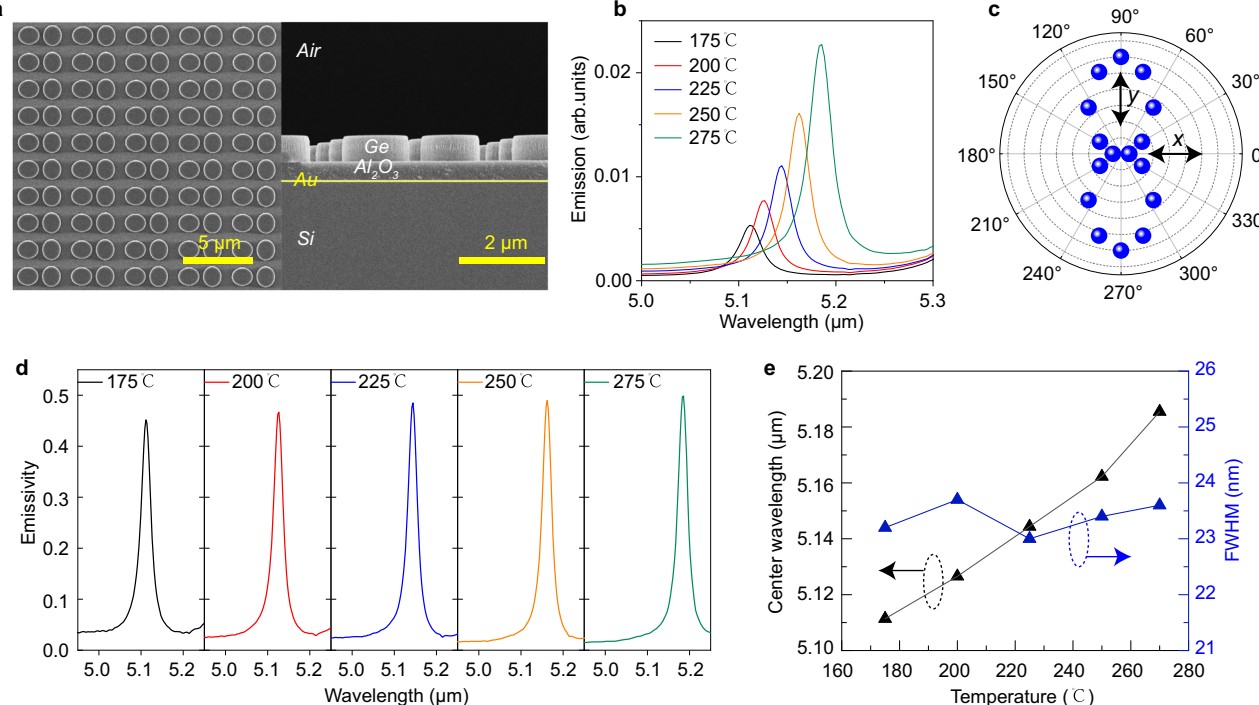

**Fig. 4 | Experimental demonstration of the thermal emissions. a** Top and $xz$ cross-sectional SEM images of the fabricated DDS structure. **b** The emission intensity of the fabricated sample at different temperatures (175, 200, 225, 250, and 275 °C) under $y$-polarization. **c** The polarization dependence of peak wavelength emission intensity at the temperature of 225 °C. **d** Normalized emission spectra at five different temperatures. **e** Extracted center wavelength and the full width at half maximum (FWHM) from (**d**).

temporal coupled mode theory (TCMT), the emissivity (absorptivity) in the whole system can be expressed as[47]:

$$E = A = 1 - R = \frac{4\gamma\gamma_0}{(\omega - \omega_0)^2 + (\gamma + \gamma_0)^2} \quad (1)$$

where $\omega$ is the frequency of incident light (equivalent to the frequency of radiations generated by the dipole oscillations from thermally excited electrons in metals), $\omega_O$ is the resonance frequency, $\gamma_O$ and $\gamma$ are the loss rate from material dissipations and radiations, respectively. In the calculations, the $Q_{rad}$ of the mode can be obtained by $Q_{rad} = \omega_O/2\gamma$; the $Q_{abs}$ can be obtained by $Q_{abs} = \omega_O/2\gamma_O$, or equivalently by the material complex refractive index[48], $Q_{abs} = n/2k$ where $n$ and $k$ are the real and imaginary part, respectively. The critical coupling condition is only satisfied for $\omega = \omega_0$, $\gamma_O = \gamma$ ($Q_{rad} = Q_{abs}$) resulting in the achievement of perfect absorption(PA)/emission(PE) and maximum field enhancement. These two Q factors can be separately controlled by the thickness of the $Al_2O_3$ buffer layer (see Fig. S5) or by the magnitude of the period-doubling perturbation. As shown in Fig. 3b, the state of PE occurs at $\Delta = 190$ nm, which is used as the optimal parameter in the final structure. Figure 3c shows the dependence of the Q-factor on the wavevector numerically calculated at the wavelength of PE. A slight decrease of the Q-factor is seen at larger wavevector arising from increased radiation losses provided by the change of $k$ as an external perturbation. The output spectrum of the thermal emitter under $y$ polarization is presented in Fig. 3d, exhibiting excellent monochromaticity with an emission linewidth of only 5.94 nm at the central wavelength of 5.131 μm. This linewidth is more than two orders of magnitude smaller than that of metamaterials-based thermal emitters[18,23]. The insets in Fig. 3d show the normalized electric field and magnetic field distributions at the peak wavelength across the $xz$ section, respectively. A significant enhancement of the field amplitude within the high refractive index Ge domain is seen, confirming that the achieved narrow-band emission results from the

coupling of thermal fluctuation to the high-Q resonance. The results of multipole decomposition (the computational details are provided in the Supplementary Information, part II) given in Fig. 3e indicate that the resonance peak is indeed dominated by a high-Q magnetic dipole. We studied the polarization characteristics of the emission output from the structure. As shown in Fig. 3f, the structure only presents high emissions in the $y$-polarization, with the spectral intensity decreases to 0 for $x$-polarization. This indicates that this nonlocal metasurface that supports flat band can emit radiations with linear polarization. This is consistent with our discussions of Fig. 1c. For the output wavelength of 5.131 μm, we calculated the 2D emission spectra of the structure towards different angles in 3D space. It is evident from the results in Fig. 3g that within an angular range of 17.5° around the normal direction, the emissivity remains near-unity. Additionally, Fig. 3h and i represent the 2D mapping results of the emissivity as functions of both the wavelength and the output angle in the $y$ and $x$ directions, respectively. For the output angle range of −17.5° to 17.5° in both directions, the resonance exhibits negligible spectral shift and relatively robust narrow linewidth. The maintenance of the emission linewidth over a large range of output angles is consistent with the results of Q-factors in Fig. 3c. These results indicate that when using a lens with a numerical aperture (NA) around 0.3 to collect the radiations from the emitter, all the energy emitted at the wavelength of 5.131 μm can be collected over a large space angle. Within this collection range, the emission does not exhibit any spectral shift, thereby increasing the emission intensity of the central wavelength and avoiding the linewidth increase induced by overlapping of multiple output wavelengths.

## Experimental demonstration
In the experiment, the DDS pattern was defined in the electron-beam resist using a 50 kV Electron Beam Lithography system. A thin layer of $Al_2O_3$ obtained by EBE and the subsequent lift-off process was used as the mask to transfer the pattern to the Ge layer using inductive

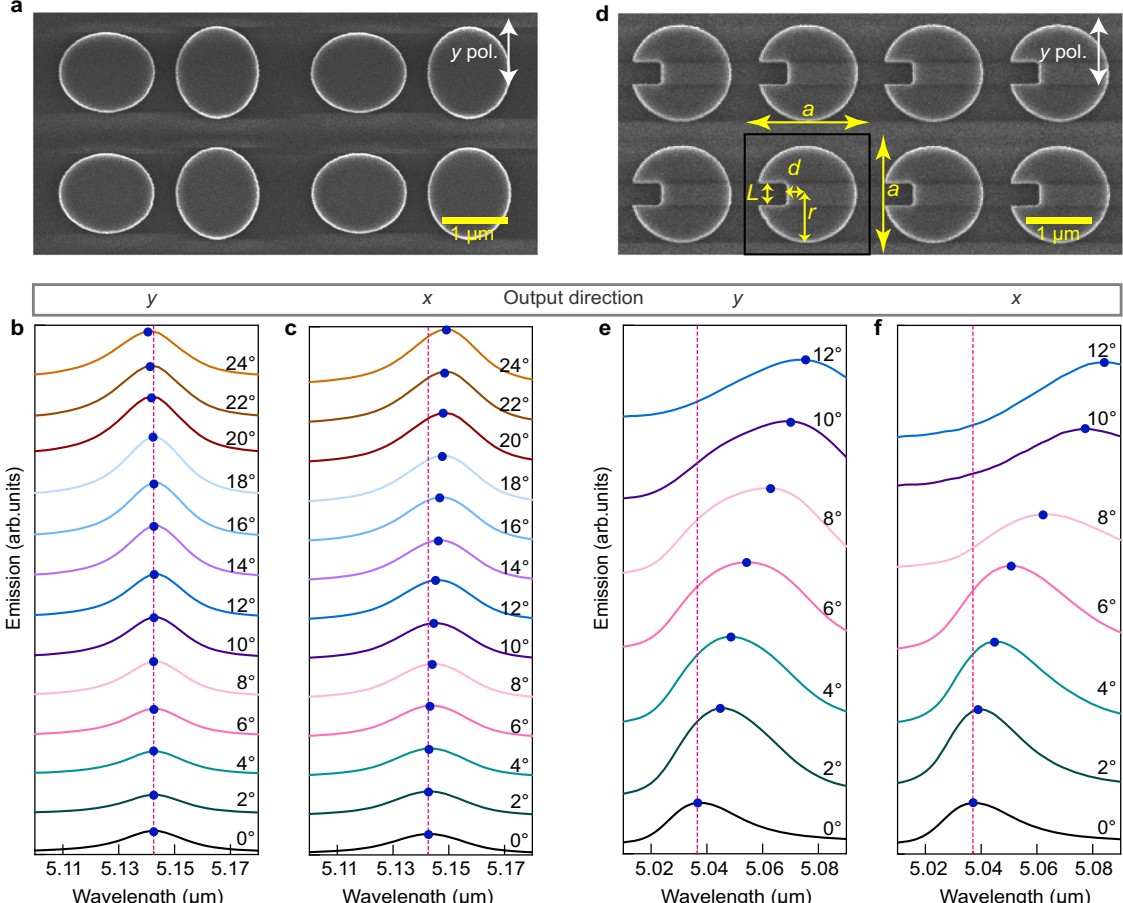

**Fig. 5 | Angular dependence of the measured thermal emissions from high-Q resonances with different dispersion characteristics. a, d** Top SEM views of the fabricated DDS and symmetry-breaking SDS (SB-SDS) thermal emitter, respectively. The white arrow indicates that all spectral measurements below were conducted under *y*-polarization. **b**–**f** Measured angle-dependent emission spectra in the *y* and *x* output direction, respectively, with an output angle scanning step of 2°. The results in (**b**) and (**c**) are from the DDS structure in (**a**) while the results in (**e**) and (**f**) are from the SB-SDS structure in (**d**).

coupled plasma-enhanced reaction ion etching. The complete fabrication process of the device can be found in Fig. S6. Figure 4a presents the top and side-view scanning electron microscope (SEM) images of the fabricated sample illustrating the excellent fabrication quality and processing precision. We employed a commercial heating adapter from Bruker to heat the sample to high temperatures, whose radiations were collected by the Fourier-transform infrared spectroscopy (FTIR) as external sources for characterization, as shown in Fig. S7. Initially, when the device surface is perpendicular to the collection direction, the large entrance window of the FTIR allows for the simultaneous entering of thermal emissions at multiple output angles into the column and being collected. Under *y* polarization, the thermal emission spectra at different temperatures (175, 200, 225, 250, and 275 °C) are shown in Fig. 4b, with the emission intensity increasing at higher temperatures. The polarization characteristics of the emission were verified by rotating the wire-grid MIR polarizer placed in the FTIR, and the results are shown in Fig. 4c. We used candle-smoked black steel plates with an area equal to the sample as the blackbody reference and normalized the emission intensities of the two samples at different temperatures to obtain the emissivity. The emissivity spectra at different temperatures are presented in Fig. 4d. A constant and ultra-narrow linewidth (FWHM = 23 nm, Q ~ 224) is found at these temperatures, which is a significant improvement over metallic metamaterials-based MIR thermal emitters. The slight decrease in the experimentally obtained emissivity compared to the numerical design is attributed to the additional scattering losses introduced by the surface and side wall roughness of the Ge disks, which will result in a mismatching between

$Q_{abs}$ and $Q_{rad}$. Figure 4e demonstrates the near linear dependence of the emission center wavelength on the temperature, extracted from the results in Fig. 4d. This is attributed to the slight change in refractive index of Ge material caused by the thermo-optic effect[49,50]. Some numerical results of the thermo-optic effect and its influence to the emission property are supplemented in Fig. S2. By referring to the previous work[49] and combining it with our results of ellipsometry measurements at room temperature, we fitted the relationship between the refractive index and the temperature. The calculated spectral shift at different temperatures closely matches the experimental results. This spectral shift actually provides a new means of fine wavelength tuning. Importantly, the output linewidth is nearly unaffected by the temperature, as indicated by the blue triangular points in Fig. 4e. This suggests that the spectral tuning by temperature is really convenient and efficient for this dielectric-based thermal emitter, which is another superior property compared to metallic metamaterials-based counterpart. The thermal dependence of the emission wavelength is beneficial to achieve precise spectral matching between the thermal emission and the absorption of the target substance, and significantly facilitates the application in NDIR sensing.

To demonstrate that the ultra-narrow bandwidth at a fixed wavelength can be achieved in a large range of output angles, i.e., the elimination of the rainbow effect, we used a three-dimensional (3D) rotating stage to characterize the emission characteristics at different angles. The schematic of measurement setup can be found in the Supplementary Information (Fig. S8). In fact, even when the sample surface is parallel to the FTIR collection entrance window, thermal radiations

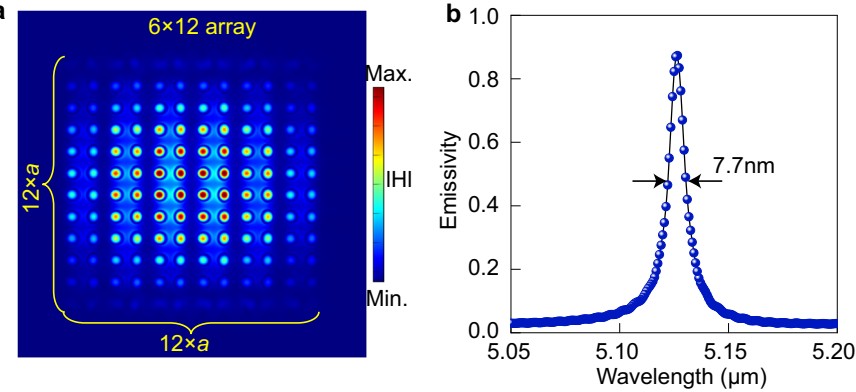

**Fig. 6 | Thermal emissions from finite periodic structures. a** Illustration the magnetic field distribution in a finite size device (6 × 12 unit cells) at the resonance wavelength. **b** Emission spectra of finite size devices in (**a**) calculated using a Gaussian beam excitation.

within a small ranger of angles can be collected, as shown by the red arrow. This is actually similar to the reciprocal process of focusing a beam in the opposite direction[42], where multiple wavevectors will be excited simultaneously. So, when we rotate the sample in different directions, the actual collection angle is still larger than the theoretical calculation. Nevertheless, as an experimental demonstration, we scanned the rotating stage in different directions at a step of 2° to obtain the angular dependence of emission spectra in either the $y$ or $x$ direction. The results at different output angles are presented in Fig. 5b, c, which exhibit a negligible shift of the resonance wavelength for the output angle up to 18° in the $y$ direction and up to 10° in the $x$ direction. This confirms that besides the ultra-narrow bandwidth of the thermal emissions, the rainbow effect can indeed be eliminated in a large range of output angles. For further larger output angles, the spectra undergo noticeable blue and red shifts, but the resonance linewidth remains almost unchanged. This is consistent with the results in Fig. 3h, i.

As a comparison to show the superior property of our flat-band design, we further designed a traditional symmetry-breaking SDS (SB-SDS) thermal emitter with the dispersion bands similar to those presented in Fig. 2a. As shown in Fig. 5d, an open slot is introduced at the edge of the Ge disk to break the symmetry. Then the SP-BIC supported by the structure will be transformed into a QBIC resonance with a high Q-factor[51]. For a fair comparison, the Q-factor of this monoatomic thermal emitter in the normal output direction was carefully designed to be equivalent to that of our DDS thermal emitter. The following geometric parameters were used: $a = 2\,\mu m$, $r = 0.8\,\mu m$, $d = 0.32\,\mu m$, $L = 0.35\,\mu m$, while the Ge layer thicknesses were identical to the DDS thermal emitter. Numerical calculation results given in Figs. S9 and S10 confirm that this thermal emitter still remains in a near-unity PE in the normal direction. We fabricated the control sample using the same process, and the top SEM view of the sample is shown in Fig. 5d. The emission spectra were also measured by rotating the sample at the step of 2° in both the $y$ and $x$ output directions. The results presented in Fig. 5e, f show that as the output angle increases, the center wavelength rapidly red-shifts, and the resonance linewidth significantly increases. Further measurement results on the emission properties from this emitter can be found in Fig. S11, including both the polarization and the temperature dependence. The linear polarization output observed in the experiment is consistent with the numerical results in Fig. S12. The Q-factors at different output angles are retrieved from the results in Fig. 5b, c and e, f, and presented in Fig. S13, with a comparison with our proposed DDS. It is clear that the DDS with the flat band dispersion not only features a rainbow-free effect, but also maintains an emission bandwidth significantly less sensitive to the output angle, compared to the SB-SDS structure. In general, for regular TEs with the rainbow effect, emissions with spectral shift in different output directions may be superimposed, causing the collected power

to be averaged. This is the main reason for the decrease in power and the increase in linewidth. Therefore, our DDS MIR thermal emitters with a flat band design exhibit superior performances.

It should be emphasized that in the practical application of thermal emitters, the use of optical lenses is necessary for more efficient power collection. Therefore, the angle-insensitive characteristic of the thermal emitter is crucial. It ensures that the overlap of spectra collected at more angles does not lead to a deterioration of the emission's temporal coherence. To characterize the collection effect of a focusing lens, we used a focused Gaussian beam in the calculation to simulate the effect of a collection lens. To reduce the computational load, we only used a finite-size device with an array of 6 × 12 DDS unit cells. By setting the waist size and area of the Gaussian beam, excitation angles covering the range of −14.5° to 14.5° (NA ~ 0.25) were achieved. Figure 6a schematically shows the magnetic field distribution at the resonance wavelength for this finite-sized emitter, confirming the same resonance mode as in the infinite-sized array. The emission spectra calculated using this finite-sized emitter and a focused Gaussian beam is presented in Fig. 6b, where the flat band effect helps maintain an ultra-narrow linewidth in the output emission spectrum even at more output angles. The slight increase in linewidth and the slight decrease in emissivity are attributed to additional scattering losses at the margin caused by finite-size effects[52], resulting in a mismatch between $Q_{abs}$ and $Q_{rad}$. However, in larger-scale metasurface devices, such scattering losses can be negligible.

With the realization of large power and narrowband thermal emissions as the objective, one can either use the proposed rainbow-free thermal emitters combined with a focusing lens to collect more radiations, or use a regular (with rainbow effect) large-area thermal emitter with a spatial filter to select the wavenumber/frequency. Since regular thermal emitters based on the QBIC[26] or QGM[27] effect exhibit strong dependence of the emission wavelength on the output angle, the spatial filter can only collect emissions within very narrow angular distributions in order to maintain the narrowband characteristics. Our estimations (see Fig. S14) show that to achieve thermal emissions with the bandwidth at the order of 10 nm, the scheme of using rainbow-free thermal emitters with a focusing lens can increase the output power by roughly two orders of magnitude.

## Discussions
Since we aim at a proof-of-concept demonstration of the new thermal emitter design in this work, all the thin films were obtained through regular EBE for the sake of experimental simplicity, which led to their amorphous states and higher absorptions. In practical applications, more advanced deposition techniques can be used to provide better material quality, which will enable a further reduction of the linewidth to even smaller values. Another thing which may raise concern in

practical applications is the spectral shift of central thermal emission wavelength due to fabrication errors. We assume an error of 20 nm, which is fully within the capability of current nanofabrication facilities, and conducted some calculations. The results shown in Fig. S15 demonstrate that with such an error, the spectral shift is only 0.04 μm, which can be easily compensated by the change of temperature according to the results in Fig. 4b.

In addition, the flat band design presented in this work can effectively achieve narrowband thermal emission with linearly polarized output. Importantly, this flat band design can be applied to more complex polarization output thermal emission designs, such as combining the intrinsic chirality of BIC in tilted structures[53] with the symmetry-breaking in the vertical direction to achieve circularly polarized output on the flat band, thereby achieving control of full polarization states while ensuring a high temporal coherence and rainbow-free characteristic of the thermal emissions.

In summary, we have shown that by combining two geometric perturbations into a square lattice, the zone-folding effect can help achieve a flat band dispersion of ultra-high Q optical states close to the Γ point. This effect has been employed to achieve ultra-narrow band and rainbow-effect-free thermal emitters operating in the MIR regime. Both numerical results and experimental demonstrations validate these two important characteristics. Compared to the previous results of thermal emissions which have high coherence but feature the rainbow effect, our results represent a further advancement in this direction since optical lenses can now be employed to collect more radiations from large-scale emitters without introducing spectral overlaps. Furthermore, the high-Q resonances like QBIC or QGM as the core of nonlocal metasurfaces require a collective interaction between a large number of periods, which suggest the use of a large sample footprint. The slow light propagation associated with the flat-band design possesses less requirement on the size of metasurfaces (see Fig. S16), which makes this rainbow-free thermal emitter more interesting. This is an important step for practical applications considering the generally low output power of thermal emitters.

## Methods

### Numerical simulation
All numerical simulations were performed based on the FEM using the commercial software of COMSOL Multiphysics. Floquet periodic boundary conditions were implemented in the lateral directions of a unit cell to define the wavevectors, while PML were employed in the $z$ direction to mimic a non-reflecting infinite domain.

### Fabrication
First, a 5 nm thick adhesion layer of titanium, a 100 nm thick Au film and another 3 nm titanium were successively deposited onto a Si substrate using electron beam evaporation. After that, two layers of 450 nm $Al_2O_3$ and 600 nm Ge were deposited onto the sample using the same equipment. Then, the 180 nm PMMA resist layer was spin-coated onto the sample and the designed structure pattern was exposed onto the PMMA layer by electron beam lithography system. After developing, 20 nm $Al_2O_3$ was evaporated onto the sample and the lift-off process was used to transfer the pattern from PMMA to $Al_2O_3$ layer. The $Al_2O_3$ pattern worked as a mask to obtain the Ge structure in a subsequent inductive coupled plasma-enhanced reaction ion etching process.

### Optical measurements
The emissivity of the fabricated sample was measured by the FTIR (Bruker Vertex 80-V) equipped with a liquid-nitrogen-cooled mercury cadmium telluride (MCT) detector. A home-made linear polarizer working in the MIR composed of one-dimensional subwavelength array of gold nanowires on a $CaF_2$ substrate was used for the polarization characterization. The emissivity measurements were carried

out by loading the sample onto an emission adapter (Bruker, A540) to heat the sample at different temperatures to induce thermal emissions. Steel plates of the same size blackened by the black soot of candles are used as the standard blackbody reference. Both the blackbody and the fabricated sample were loaded onto the heating plate. When the blackbody and the sample are heated to above 100 °C, the emitted power is sent to the FTIR system and detected by MCT detector.

## Data availability
All the data in this study are provided within the paper and its supplementary information.

## Code availability
All the codes that support the findings of this study are available from the corresponding author upon request.

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

## Acknowledgements
This work is sponsored by the National Natural Science Foundation of China (12274269, 11974221, 12192254 and 92250304) and the National Key Research and Development Program of China (2022YFA1404800).

## Author contributions
Z.H. conceived the idea and supervised the project. K.S. performed all the experiments. K.S. and Z.H. analyzed the data and prepared the manuscript. L.H. co-supervised the project and participated in the numerical calculations. Y.C. participated in the discussion. All authors participated in the manuscript writing.

## Competing interests
The authors declare no competing interests.
