## [Peer Review File · Nature Communications]

REVIEWER COMMENTS

Reviewer #1 (Remarks to the Author):

Thermal emitters are viable for various practical applications. However, there are several works to improve the efficiency of thermal emitters by integrating with photonic cavities and 2D materials. In the current work, authors have employed the flat band concept, which is not new. By introducing the geometrical asymmetry and periodicity, band folding can be realized which can give a flat band at the gamma point. However, in the current work, it is still unclear why this structure exhibits a flat band for a large range of k-values, which is eventually helping to achieve angle-independent emissivity enhancement. I suggest a major revision which can bring more clarity

1. In Fig. 1b the evolution to achieve the final structure is not illustrated properly. The authors should illustrate it more carefully. Labelling of '3' and '4' are misleading.
2. Also, in the final structure, indeed the symmetry along the y-direction is broken but the $\text{real}(E_y)$ component does not exhibit a significantly different modal profile compared to other structures. Then how just by looking at the modal profile of the $\text{real}(E_y)$ of the final structure, authors claim it as a BIC mode. More justifications are required.
3. In Fig. 2A where is the GMR mode? Is it overlapped with BIC mode?
4. Also, in Fig. 2B in the T-Y direction, authors must label the QGM and GMR mode and two GMRs at T-X. And what is the origin of two GMR modes in the T-X? It is not clear. There has to be one GMR mode only in this direction.
5. What is the underlying physics behind the flat band in DDS? How does the slow light feature emerge in DDS? Authors must highlight this, as it is extremely important.
6. In Fig 3, authors must highlight the definition of Δ somewhere in the inset.
7. In Fig. 3c, is it measured Q factor or simulated?
8. Authors must compare their results with the existing state-of-the-art other work and tabulate it properly. There are various ways to achieve flat band at the gamma point, for example, photonic crystal. How is their approach superior to these approaches? All these discussions have to be thereto clearly show the novelty of their work.

All the above revisions, the manuscript could be considered for publication at Nature Com.

Reviewer #2 (Remarks to the Author):

With interest I have read the present manuscript, showing nice thermal emission experiments. Starting with a few broader comments:

- Overall, I think the distinction between polarization and direction of rotation could be made clearer, and often discussion of polarization appears to be missing. For example, it would be interesting to see fig. 3F for both polarizations, or ideally even full polarimetry for this metasurface. As the authors rightfully point out, metasurfaces give control over the emitted polarization state, so it would be fitting to incorporate this into the manuscript as well.
- On that topic, could the authors discuss how to incorporate more intricate polarization states into the flat band approach?
- Fig. 5 could be a color plot, similar to Fig. 3F. It is also not apparent what the polarization is here, or if x/y refer to polarization (I think this is the direction of the angle);
- The discussion regarding finite structures, the wide angular spread, and the collection optics is interesting, but it would be very useful if this discussion could be extended: could you compare a large metasurface of area A that emits truly into one narrow cone with solid angle Ω (with large spatial coherence) with a smaller metasurface from which emission is collected by a lens of area A? Conservation of etendue is important here, since this light is not coherent it cannot be focused into a collimated beam: consider the same solid angle Ω for both systems.
- It appears that you either have dispersion in frequency or in angle: can you briefly discuss this trade-off? To emit into a single direction for a single frequency, what would be needed (is it a bulls eye grating?)?
- The content is here to make this an impactful publication, but the presentation is not really there yet. I think the first figure could use schematics illustrating the "regular" metasurfaces with parabolic dispersion and the flatband structure, including suggested setups of large area vs smaller area + collimating lens. Additionally, I would try and make the figures a little cleaner, which can partially be achieved by not using jet/rainbow color scales.

Smaller comments:

- Line 47: I would say that thermal radiation is generally "unpolarized" rather than "polarization independent".
- Fig. 3D: the $|E|$ and $|H|$ are not really legible on top of the color legend.
- Fig. 4B/D: these seem redundant.
- Fig. 5, line 319: "depdent" should be "dependent"

Reviewer #3 (Remarks to the Author):

The paper is an interesting work combining BIC with for thermal emission control, namely wavelength. Few questions and comments:

- The authors discuss the need for narrowband emission but it is not very clear the benefits. In other words usually one is interested to control a wideband range of emissions from the source (to save the efficiency)
- The authors discuss other alternatives like to have two beams and use filter to remove one of them, but

then one can make proper design with only one beam direction, please see below papers,

S. Inampudi et al, "Unidirectional thermal radiation from SiC metasurface," JOSA B.

S. Inampudi et al, "Tunable wideband-directive thermal emission from SiC surface using bundled graphene sheets," Phys. Rev. B.

- It is recommended the authors may include them in references due to relevance

- Talking about efficiency, the BIC is very high Q, then even small loss can result very low efficiency. Perhaps better the authors show results, without normalization

- Isn't the design polarization sensitive? Then for a thermal heat 50% of the power may not be manipulated properly?

- How sensitive fabrication is to the required design? Given BIC what is tolerance in fabrication within the elements?

The reviewer can recommend publication after careful addressing the above comments.

Response to Reviewer Comments

Reviewer #1:

Thermal emitters are viable for various practical applications. However, there are several works to improve the efficiency of thermal emitters by integrating with photonic cavities and 2D materials. In the current work, authors have employed the flat band concept, which is not new. By introducing the geometrical asymmetry and periodicity, band folding can be realized which can give a flat band at the gamma point. However, in the current work, it is still unclear why this structure exhibits a flat band for a large range of k -values, which is eventually helping to achieve angle-independent emissivity enhancement. I suggest a major revision which can bring more clarity.

Response: We are very grateful to the reviewer for giving us the opportunity to revise and improve the quality of our manuscript. We completely agree with the reviewer that the flat band effect itself is not new, even though how to realize this effect efficiently is still an active topic in photonics. There are many methods to achieve the flat band effect. However, it is still very challenging to achieve both flat band effect and high Q-factor simultaneously through a simple structure. For example, the widely studied Moiré structure in photonic crystals has been used to achieve large-wavenumber flat band effect, but its complex manufacturing requires extremely high-dimensional accuracy of the structure. Another way of achieving flat band is to engineer the energy-momentum dispersion relationship through harnessing symmetry-broken gratings, but the significant breaking results in relatively low Q-factors. Plasmonic BIC is also used to achieve angle insensitive flat band effects, but the high inherent loss of metals makes it impossible to achieve high-Q characteristics. These approaches have been discussed in the revised manuscript now.

However, to the best of our knowledge, by harnessing this effect to achieve thermal emissions with simultaneous rainbow-free and narrowband characteristics has never been reported before. It should be emphasized that these two features are important for thermal emitters in practical applications given the generally weak output from thermal emitters. Thus, it is necessary to use a focusing component like lenses to collect more radiations from large-area thermal emitters. In this respect, it is important to eliminate the rainbow effect, otherwise multiple-resonances with different frequencies corresponding to different wavenumbers will be collected by the lens and overlap with each other, resulting in a degradation of the temporal coherence. A new figure of Fig. 1(a) has been added to the manuscript to highlight the importance of eliminating the rainbow effect. We believe that realizing a rainbow-free and narrowband thermal emitter represents a solid step for the development of thermal emitters for practical applications.

To make it clearer why this structure exhibits a flat band for a large range of k -values, we have made careful revisions. The essential part to achieve this effect is the strong and steered coupling between the quasi-guided modes (QGM) and the originally existing guided-mode resonances (GMR) in both the k_x and k_y directions. Here we decide to use the term of QGM, although it is also a special type of GMR. We have highlighted the difference between them in the revised manuscript using "As an **exceptional type of guided-mode resonance (GMR)**, the **QGMs** not only exhibit perturbation-dependent Q-factors, but also advantageously feature high robustness of the Q-factor against the frequency/wavenumber over the QBICs". To host the two modes of QGM and GMR, the original photonic lattice should fulfill the following requirements. At small wavevectors close to the Γ point, GMR bands should be supported along both ΓX and ΓY directions. However, each GMR band extends across the light line at larger wavenumbers close to the X and Y points, where the GMR switches to GM. Particularly, the dispersion band along the XM direction continuing from the X point is all below the light line. After the period-doubling perturbation, the GM part at larger wavenumbers along the ΓX direction will also be folded back to the Γ point. At the same time, the dispersion band of the GM along the XM direction in the original FBZ is folded to the ΓY direction. Then we have two bands of GMR and QGM along both the k_x and k_y directions in the distorted lattice now. The strong coupling between the two bands along each direction gives rise to the splitting of them into two new bands. Importantly, the coupling strength and the slopes of the two new bands can be adjusted by changing the geometry like the center-to-center distance " $a-\delta$ " or height h of the two disks (see Fig. S4). The flat-band behavior can then be achieved in one of the two new bands.

We have combined and re-ordered the results in the original Fig. 1 and Fig. 2 to elucidate the working principal. We first present the requirement of the original square lattice. It should support GMR at smaller wavenumbers and GM at large

wavenumbers along both ΓX and ΓY directions. The mode should be complete GM along the XM direction. We have used the results in Fig. 2(a) as an example to illustrate these properties. After some discussions on the requirement of the period-doubling perturbation by combing processes (1) and (2) shown in Fig. 1(c), the folding of those GMs to the Γ point to form QGMs and the coupling between the QGMs and originally existing GMRs are discussed. Fig. 2(b) presents the folding effect and the band-splitting due to the strong coupling between the GMRs and the QGMs. The coupling can be steered by the geometry to achieve the flat band effect in one of the two new bands and this point is discussed in the Supplementary Information (Fig. S4).

We have added more discussions on the mode properties of the GMRs and GMs in the original square lattice of disk array as follows:

"... The calculated dispersion band of the supported modes in this lattice is presented in Fig. 2(a). At small wavevectors close to the Γ point, GMR bands are supported along both ΓX and ΓY directions. However, each GMR band extends across the light line at larger wavenumbers close to the X and Y points, where the GMRs switches to GMs. Particularly, the dispersion band along the XM direction continuing from the X point is all below the light line (see the red line in Fig. 2(a)), indicating that it is a complete GM band. Due to the C_4 symmetry of the lattice, the ΓX and ΓY directions have the same band structures. In addition, the GMR and GM bands have the same frequency at the two high-symmetry points of X and Y... "

Other details of the period-doubling perturbation and the strong coupling between the GMRs and QGMs along both the k_x and k_y directions can be found in our subsequent responses. We hope that with the above revisions, the underlying physics why the structure supports the flat-band effect has been clarified and it is easier for the readers to follow now.

1. In Fig. 1b the evolution to achieve the final structure is not illustrated properly. The authors should illustrate it more carefully. Labelling of '3' and '4' are misleading.

Response: We thank the reviewer for this valuable suggestion. We have made changes to Fig. 1 by removing the original labels (3) and (4), as well as adding new illustrative arrows, to better illustrate the evolution process of the structure. In addition, we have provided a more detailed description of Fig.1.

"For the next step, it is required to introduce a period-doubling perturbation into the square lattice to manipulate the FBZ along the k_x direction. There are two ways of triggering this perturbation. One is to change the circular disks to elliptical ones and rotate every second column of them by 90° without changing the center-to-center distance "a", as shown by operation (1) in Fig. 1(c). The other, as shown by operation (2), is to change the center-to-center distance to "a- δ " between adjacent disks without changing the disk geometry."

" To break the C_2 symmetry of the structure, we combine these two operations by introducing elliptical shapes with different orientations and changing the center-to-center distance "a- δ " at the same time. The final design of the structure is shown in the bottom right panel of Fig. 1(c). "

Fig.1 | (a) Schematic diagram illustrates the spectral response of high-Q nonlocal metasurfaces with different dispersion behaviors when excited by a focused beam. (b) Artistic rendering of the thermal emitter, where the red output beam represents the thermal radiation at the QBIC resonance. (c) illustrates the evolution process from the traditional SDS to the final DDS structure, with red and blue block diagrams representing the real parts of E_x and E_y and the dashed line representing the axis of symmetry in the polarization direction. (d) The corresponding shrinking of the FBZ due to the introduction of the perturbation.

2. Also, in the final structure, indeed the symmetry along the y -direction is broken but the real(E_y) component does not exhibit a significantly different modal profile compared to other structures. Then how just by looking at the modal profile of the real (E_y) of the final structure, authors claim it as a BIC mode. More justifications are required.

Response: We thank the reviewer for this valuable suggestion. We agree that the difference between the modes in those structures in Fig. 1(c) is not so significant because the geometric perturbation is weak in order to maintain a relatively high Q -factor. We note the difference between BIC and QBIC mode can be better seen from the far-field polarization maps. It is known (PRL 113, 257401(2014)) that BICs are vortex centers in the polarization directions of far-field radiation. For the DDS implemented by either process (1) or (2), the structure still possesses C_2 symmetry. The far-field polarization vector cannot be defined and still manifests as a vortex center carrying +1 topological charge, as shown in Fig. S3, which is a characteristic of BIC with infinite Q_{rad} factors. For the final structure combining processes (1) and (2), there is no longer an axis of symmetry along the y -direction, and real (E_y) component doesn't have a perfectly anti-symmetric modal distribution due to symmetry breaking. Therefore, the mode can be coupled with external plane waves. The far-field polarization exhibits linear polarization characteristics in the y -direction, as shown in Fig. S3(d), which is completely consistent with the modal discussion in Fig.1(c). In order to make the BIC/QBIC modality clearer for readers, we have provided more supplements in Fig. S3 as well as some descriptions in the main text.

“...It is worth noting that although both methods can reach the period doubling in the x -direction, the evolved DDS still retains its C_2 symmetry, either around the center between two disks or around the center of each disk. In both cases, the real

parts of both electric field components still exhibit a perfectly anti-symmetric distribution along the structural symmetric axis (dashed line) in their respective polarization direction. Thus, these modes cannot couple to external plane waves which have even distributions of the electric field. For the DDS implemented by either process (1) or (2), the far-field polarization vector cannot be defined and still manifests as a vortex center carrying +1 topological charge, as shown in Fig. S3, suggesting the nature of BIC with infinite Q_{rad} factors. To break the C_2 symmetry of the structure, we combine these two operations by introducing elliptical shapes with different orientations and changing the center-to-center distance “ $a-\delta$ ” at the same time. The final design of the structure is shown in the bottom right panel of Fig. 1(c). In this case, although the E_x field distribution still maintains a perfect anti-symmetry with respect to the x -axis, the structure is no longer symmetrical along the y direction, and the E_y mode distribution is disrupted, allowing for coupling with y -polarized plane waves. This can be confirmed by its far-field polarization (see Fig. S3(d)), indicating the conversion of BIC into QBIC. It exhibits a nonlocal resonant mode with a high yet finite Q-factor”.

Fig. S3 | (a) (b), (c), and (d) represent the far-field polarization of different structures in Fig.1(c). (e) The far field polarization of symmetry breaking SDS.

3. In Fig. 2A where is the GMR mode? Is it overlapped with BIC mode?

Response: We are thankful to the reviewer for the careful reading and raising this good question. In periodic photonic structures, the GMR is supported in the continuum region. As discussed in the literature (*Optica* **9**, 1353 (2022)), the symmetry-protected BIC comes from the coupling between degenerate GMR modes, so it is a special case located at the Γ point but embedded within the band of GMR.

To address this comment, we have updated both Fig. 2(a) and Fig. 2(b) with more labels now to distinguish between GMR and GM now. The special point of BIC/QBIC is also labeled at the Γ point to show its position.

Fig.2 | (a) shows the dispersion band supported by SDS along different directions within the FBZ, respectively, and (b) presents the band in the new FBZ after the transformation from SDS to DDS. The insets show the amplitude distribution of the magnetic field and the vector distribution of the electric field. **The black circle represents the wavevector region of strong mode coupling.** (c) and (d) are the corresponding group velocities v_g close to the Γ point for modes supported by SDS and DDS, respectively.

4. Also, in Fig. 2B in the Γ -Y direction, authors must label the QGM and GMR mode and two GMRs at Γ -X. And what is the origin of two GMR modes in the Γ -X? It is not clear. There has to be one GMR mode only in this direction.

Response: We appreciate the careful reading and a valuable suggestion by the reviewer. As we responded to the previous question, we have updated both Fig. 2(a) and Fig. 2(b) with more labels to reveal the properties of different modes. The two bands presented in Fig. 2(b) in both Γ -X' and Γ -Y directions are the consequences of the band splitting from the strong coupling between GMR and QGM at the presence of the period-doubling perturbation. It is this strong coupling which is steered by the geometric changes that induces the flat band effect.

5. What is the underlying physics behind the flat band in DDS? How does the slow light feature emerge in DDS? Authors must highlight this, as it is extremely important.

Response: We highly appreciate the constructive suggestion from the reviewer. As we responded to question 1, the essential part to achieve the flat band effect with overall high Q-factors is the strong and steered coupling between the quasi-guided modes (QGM) and originally existing guided-mode resonance (GMR) in both the k_x and k_y directions.

To address the reviewer's comment, we repeat some of our previous responses here. To host the two modes of QGM and GMR, the original photonic lattice should fulfill the following requirement. At small wavevectors close to the Γ point, GMR bands should be supported along both ΓX and ΓY directions. However, each GMR band extends across the light line for larger wavenumbers close to the X and Y points, where the GMR switches to GM. Particularly, the dispersion band along the XM direction continuing from the X point is all below the light line. After the period-doubling perturbation, the GM part at larger wavenumbers along the ΓX direction will also be folded back to the Γ point. At the same time, the dispersion band of the GM along the XM direction in the original FBZ is folded to the ΓY direction. Then we have two bands of GMR and QGM along both the k_x and k_y directions in the distorted lattice now. The strong coupling between the two bands along each direction gives rise to the splitting of them into two new bands. Importantly, the coupling strength and the slopes of the two new bands can be

adjusted by changing the geometry like the center-to-center distance “ $a-\delta$ ” or height h of the two disks (see Fig. S4). The flat-band behavior can then be achieved in one of the two new bands.

We have provided more detailed descriptions in the abstract and main text in the revised manuscript now.

In the abstract:

"... As a result of the first Brillouin zone halving, the guided modes will be folded to the Γ point and interact with originally existing guided-mode resonances to form a flat band of dispersion with overall high Q..."

In the main text:

"...This transformation folds the dispersion band of the GM along the XM direction in the original FBZ to the Γ Y direction (i.e. QGM now)⁴⁵ in the new lattice, as shown by the red band in the Γ Y direction in Fig. S4(a). Furthermore, the GM part at larger wavenumbers along the Γ X direction will also be folded back to the Γ point, as shown in the folded band in Fig. 2(b). Therefore, there will be two dispersion bands in both the k_x and k_y directions in the new FBZ, one is QGM and the other is the originally existing GMR. The strong coupling between the two bands along each direction gives rise to the splitting of them into two new bands. A clear avoided-crossing behavior is seen in Fig. 2(b) along the Γ Y direction, which is a signature of the strong-coupling effect. The field distributions at the Γ point for the two new bands are presented in the inset of Fig. 2(b), which clearly shows that they retain the modes of the original BIC and GM. Importantly, the coupling strength and the slopes of the two new bands can be adjusted by changing the geometry like the center-to-center distance “ $a-\delta$ ” or height h of the two disks (see Fig. S4). The flat-band behavior can then be achieved in one of the two new bands. Through careful design, we have adopted the following perturbation parameters: $\delta=0.25\ \mu\text{m}$, elliptical disk long axis $A=1.54\ \mu\text{m}$ and short axis $B=1.35\ \mu\text{m}$. The new band in the upper part of Fig. 2(b) shows a pronounced flat band behavior in a wide wavevector range. We further solved for the group velocity using $v_g=d\omega/dk$ close to the Γ point for the GMR band both in the SDS and the flat band in the DDS, and the results are shown in Figs. 2(C) and (D), respectively. Clearly, for the GMR band in the SDS, the group velocity rapidly increases at larger wavevectors, reaching $0.003(c/\pi)$ at a wavevector of $0.1\pi/a$. In contrast, for the DDS, v_g remains almost constant and close to 0. In the latter section, we show that such an ultra-low group velocity plays a crucial role in achieving narrowband thermal emissions free from the rainbow effect."

6. In Fig 3, authors must highlight the definition of Δ somewhere in the inset.

Response: We thank the reviewer for this valuable suggestion. Accordingly, we have changed the “ Δ ” in Figs. 3(a) & (b) to “ $A-B$ ”.

7. In Fig. 3c, is it measured Q factor or simulated?

Response: We thank the reviewer for careful reading and bringing up this question. Fig. 3(c) shows the simulated values, which predict the changes in the Q-factor under different wavevectors. The maintaining of high Q values at different wavevectors is important and the numerical results serve as a guideline for the subsequent experimental work. We have changed the description both in the main text and in the caption of Fig. 3(c) to make it clearer.

In addition, we have added the measurement results of Q-factor at different angles, as shown in the Supplementary Information of Fig. S13, and added the following corresponding descriptions.

“Fig. S13 shows the Q-factors extracted from Figs.5 (b), (c), (e), and (f). For DDS that supports the flat band, its Q-factor remains almost unchanged as the wavevector increases. Due to the simultaneous collection of multi-angle thermal emissions by the FTIR system, for SDS that supports the rainbow bands, it is seen that the Q-factor rapidly decreases with the increase of output angle.”

Fig. S13 | The Q-factors of the two structures extracted from Figs.5 (b), (c), (e), and (f) at different output angles. The black and blue curves represent the results in the ΓY direction, the red curve represents the results in the ΓX direction, and the green curve represents the results in the $\Gamma X'$ direction.

We have also added the sentences in the main text to refer the readers to the comparison:

"...The Q-factors at different output angles are retrieved from the results in Figs. 5(b)-(c) and (e)-(f), and presented in Fig. S13, with a comparison with our proposed DDS. It is clear that the DDS with the flat band dispersion not only features a rainbow-free effect, but also maintains a narrow emission bandwidth significantly less sensitive to the output angle, compared to the SB-SDS structure..."

8. Authors must compare their results with the existing state-of-the-art other work and tabulate it properly. There are various ways to achieve flat band at the gamma point, for example, photonic crystal. How is their approach superior to these approaches? All these discussions have to be there to clearly show the novelty of their work.

Response: We greatly appreciate the reviewer's constructive suggestion. In our revised manuscript, we provide a detailed table to compare our work with previous research results in terms of fabrication accuracy requirements, Q-factor, and spectral tunability. This table emphasizes the superior characteristics of our proposed method in various aspects.

The added content in the revised manuscript is as follows:

"The flat band design has been extensively studied in fundamental optical sciences due to the large compatibility of all the spatial components at the same frequency under wide-angle illumination, as well as the associated slow light effect which can greatly increase the interaction time between light and matter. However, a simultaneous realization of flat band dispersion and high Q-factor can be challenging. Table 1 presents the compared performance of flat band designs in periodic photonic structures achieved by different approaches. The use of moiré structures to achieve flat bands is conceptually appealing^{35,39,40}, with some reported experimental confirmation of low-threshold micro-lasers⁴¹. However, the complex fabrication process and uncontrolled radiation characteristics make it difficult to be widely adopted in practical applications. Nguyen et al. attempted to manipulate energy-momentum dispersion relationships through symmetry breaking^{36,42}, but the large breaking makes it difficult to maintain the high-Q property. Plasmonic BICs have been demonstrated to achieve angle-insensitive flat band effects^{37,43}, but the intrinsic losses of metals make it challenging to achieve high Q-factors. Some researchers have used high-order Mie resonances in all-dielectric metasurfaces to achieve wavefront shaping³⁸, thereby obtaining high numerical aperture lenses. However, the Q-factor obtained by Mie resonance is also relatively low. Therefore, it still remains an unsolved challenge to achieve both high Q-factors and flat band effects within a wide range of wavenumbers."

Table 1|Comparison of performance of flat dispersion bands achieved by different methods

Structures	Fabrication accuracy requirements	Q-factor	Q or Wavelength tunability	Ref.
Moiré photonic crystal	High	High	Low	35
Symmetry breaking grating	Low	Low	High	39
Plasmonic BICs	Low	Low	High	41
Mie-resonant metasurfaces	Low	Low	High	43
Distorted photonic lattices	Low	High	High	This work

All the above revisions, the manuscript could be considered for publication at Nature Com.

Response: We are grateful for the positive feedback from the reviewer and for considering the manuscript suitable for publication in Nature Communications. We sincerely appreciate the time and effort invested by the reviewer in reviewing our work. We hope that our response and revisions can address all the concerns.

Reviewer #2:

With interest I have read the present manuscript, showing nice thermal emission experiments. Starting with a few broader comments:

Response: We greatly appreciate the kind words and the time spent in reviewing our manuscript. We are delighted to hear that the reviewer found the thermal emission results presented in our manuscript "nice". The positive feedback is encouraging, and we appreciate the professional knowledge and important suggestions from the reviewer to help us improve our work.

1. Overall, I think the distinction between polarization and direction of rotation could be made clearer, and often discussion of polarization appears to be missing. For example, it would be interesting to see fig. 3F for both polarizations, or ideally even full polarimetry for this metasurface. As the authors rightfully point out, metasurfaces give control over the emitted polarization state, so it would be fitting to incorporate this into the manuscript as well. On that topic, could the authors discuss how to incorporate more intricate polarization states into the flat band approach?

Response: We thank the reviewer for this excellent suggestion. We have update Fig. 3 with a new Fig. 3(f) to present the full polarization of thermal emitter metasurfaces in the revised manuscript. As we discussed about the process flow in Fig. 1(c), the final structure with a breaking of C_2 symmetry supports a mode with the E_x field maintaining a perfect anti-symmetry with respect to the x -axis. The slight disruption of E_y mode distribution from anti-symmetry allows for its coupling with linearly polarized plane waves in the y direction. Then according to Kirchhoff's law of thermal radiation, this structure presents a thermal emission from the reciprocal process with the y -polarization.

In response, we have supplemented and discussed the polarization property of our symmetry-breaking SDS.

“...We studied the polarization characteristics of the emission output from the structure. As shown in Fig. 3(f), the structure only presents high emissions in the y -polarization, with the spectral intensity decreases to 0 for x -polarization. This indicates that this nonlocal metasurface that supports flat band can emit radiations with linear polarization. This is completely consistent with our discussions of Fig.1(c)...”.

We have added the following paragraph in the “Discussions and Conclusion” section to discuss how to incorporate more intricate polarization states into the flat band approach:

“In addition, the flat band design presented in this work can effectively achieve narrowband thermal emission with linearly polarized output. Importantly, this flat band design can be applied to more complex polarization output thermal emission designs, such as combining the intrinsic chirality of BIC in tilted structures⁵³ with the symmetry-breaking in the vertical direction to achieve circularly polarized output on the flat band, thereby achieving control of full polarization states while ensuring a high temporal coherence and rainbow-free characteristic of the thermal emissions. ”

[53] Chen, Y. *et al.* Observation of intrinsic chiral bound states in the continuum. *Nature* **613**, 474–478 (2023).

Fig.3 | (a) Dependence of the Q-factor on the difference $A-B$ (Δ) between the long and short axes of the elliptical disks. The blue curve corresponds to the results with evaporated amorphous materials, while the black curve is for the results where the losses of all materials are neglected. (b) The relationship between Q_{total} (blue), Q_{abs} (red), and Q_{rad} (black) at different values of Δ . Q_{rad} decreases with increasing Δ . Perfect emission is achieved when $Q_{\text{abs}}=Q_{\text{rad}}$, as shown by the yellow star symbol, which corresponds to $\Delta=190$ nm used in the final thermal emitter. (c) The **calculated** Q-factor at the perfect emission wavelength as a function of wavevectors along the ΓY and $\Gamma X'$ directions. (d) and (e) represent the emission spectrum and multipole decomposition of the resonance in the normal output direction, respectively, indicating the ultra-narrow linewidth and the main contribution from the magnetic dipole. The insets in (d) illustrates the distribution and amplitude of the electric and magnetic fields at the peak wavelength. (f) **The emissivity at the emission peak in (d) along different directions is used to indicate the y-polarized property.** (g) The emissivity at the normal output wavelength ($5.131\mu\text{m}$) as a function of output angle in 3D space. (h) and (i) represent a 2D mapping of the emissivity as a function of the output angle and wavelength in both the y and x directions, respectively.

We have also added the following results to the Supplementary Information to discuss about the polarization property of the thermal emission output from the regular symmetry-breaking SDS structure, whose main emission spectrum is presented in Fig. 5 in the main text for comparison.

“Fig. S12 shows the dependence of emissivity on the polarization direction calculated at the center emission wavelength from the SB-SDS based on the QBIC modes. The results indicate that the output characteristics of this thermal emitter is linearly polarized in the y -direction, with zero emissivity under x -polarization.”

Fig.S12 | Polarization dependence of the emission intensity from the SB-SDS.

2. Fig. 5 could be a color plot, similar to Fig. 3F. It is also not apparent what the polarization is here, or if x/y refer to polarization (I think this is the direction of the angle);

Response: We appreciate the careful reading and a valuable suggestion by the reviewer. Fig. 5 present the emission spectra collected at multiple output angles. The peak emission wavelength from the same structure in the normal output direction is annotated to highlight the spectral shift and the angular dependence. These results will help identify the change of central emission wavelength with respect to the output angle in two directions, as well as the change in the emission bandwidth during the whole rotation process. We find these features cannot be fully presented in 2D color images, so we hope the reviewer can allow us to continue using current plotting. For better presentation of the results, we have updated Figs. 5(a) and (b) with new labels to indicate the polarization direct. We have also updated the caption of Fig. 5 to highlight that "The white arrow indicates that all spectral measurements below were conducted under y-polarization." and that the presented results in Figs. 5(b)~(c) and (e)~(f) are "Measured angle-dependent emission spectra in the y and x output direction".

Fig. 5 | (a) and (d) Top SEM views of the fabricated DDS and symmetry-breaking SDS (SB-SDS) thermal emitter, respectively. The white arrow indicates that all spectral measurements below were conducted under *y*-polarization. (b)–(f) Measured angle-dependent emission spectra in the *y* and *x* output direction, respectively, with an output angle scanning step of 2°. The results in (b) and (c) are from the DDS structure in (a) while the results in (e) and (f) are from the SB-SDS structure in (d).

3. The discussion regarding finite structures, the wide angular spread, and the collection optics is interesting, but it would be very useful if this discussion could be extended: could you compare a large metasurface of area *A* that emits truly into one narrow cone with solid angle Ω (with large spatial coherence) with a smaller metasurface from which emission is collected by a lens of area *A*? Conservation of etendue is important here, since this light is not coherent it cannot be focused into a collimated beam: consider the same solid angle Ω for both systems.

Response: We thank the reviewer for a really good question regarding the use of the rainbow-free thermal emitters for practical applications with a focusing lens to help collect more power from the emitter. We have conducted rough estimations based on our previous results on the level of emission linewidth that can be achieved experimentally based on non-local metasurfaces (*Phys. Rev. Appl.* **20**, 024033 (2023)). The following results on how to make the estimations have been added to the Supplementary Information:

"We compare the collected level of powers of narrowband thermal emissions from two different types of thermal emitters. One is the regular type of thermal emitter with rainbow effect based on the QBIC or QGMs, and the other is the rainbow-free thermal emitter proposed in this work. For the former case, a spatial filter like a hole in an opaque screen is required to select the spectral component. Due to the steep dispersion of the QBIC or QGMs, the resonance frequency is highly dependent on the output angle. As a result, the hole should be very small to ensure the final output with an adequately narrow band. For the latter case, a focusing lens can be used to collect the radiations within a large angle (We keep the angle to be 17.5° based on the results in the main text). For a fair comparison, the size of the emitter in the first case is assumed to have the same area with the size of the lens in the second case. The two schemes with the respective output characteristics are shown in Fig. S14.

One can obtain the emitted power from the first case as follows:

$$P_1 = \frac{\Delta\theta}{180^\circ} \eta \pi (0.5D_1)^2 \quad (10)$$

where $\Delta\theta$ is the full-width at half maximum in the angular distribution of the thermal emissions at a specific wavelength, η is the coefficient of thermal emissions from unit area size. From the results in our previous studies^{8,9}, it is known that $\Delta\theta$ is very small and its value is dependent on the objective emission bandwidth and which part of the steep dispersion is used. For the thermal emission with the bandwidth at the order of 10 nm, a value of 0.43° is used for rough estimation based on our previous results⁹.

For the second case, the collected power is:

$$P_2 = \eta S_{eff} = \eta \pi (0.5D_1 - L_0 \tan(17.5^\circ))^2 \quad (11)$$

where S_{eff} is the effective area from which the thermal radiations can be collected within the angle of 17.5°, to ensure that the flat band effect works. L_0 is the distance between the focusing lens and the sample. One can see that the collected power in the second case is dependent on the value of L_0 . To make the estimations practical, we assume that $L_0 = 0.5 * D_1$, both can be assumed at the length scale of millimeters.

With above assumptions, one can calculate that the power ratio from the two cases:

$$\frac{P_2}{P_1} = \frac{(1 - \tan(17.5^\circ))^2}{180} = 168 \quad (12)$$

This estimation result suggests that our scheme of combining the rainbow-free thermal emitter with a focusing lens can increase the power output by two orders of magnitude compared to the case of using large-area rainbow-type thermal emitters with a spatial filter. "

Fig. S16| (a) and (b) Two schemes of obtaining large-power thermal emissions with high temporal coherence. (c) and (d) The spectral properties of the two types of thermal emitters at a function of wavevector.

Correspondingly, we have summarized the estimation results and added the following discussions to the main text:

"With the realization of large power and narrow-band thermal emissions as the objective, one can either use the proposed rainbow-free thermal emitters combined with a focusing lens to collect more radiations, or use a regular (with rainbow effect) large-area thermal emitter with a spatial filter to select the wavenumber/frequency. Since regular thermal emitters based on the QBIC²⁶ or QGM²⁷ effect have strong dependence of the emission wavelength on the output angle, the spatial filter can only collect emissions within a very narrow angular distributions in order to maintain the narrow-band characteristics. Our estimations (see Fig. S14) show that to achieve thermal emissions with the bandwidth at the order of 10 nm, the scheme of using rainbow-free thermal emitters with a focusing lens can increase the output power by roughly two orders of magnitude. "

4. It appears that you either have dispersion in frequency or in angle: can you briefly discuss this trade-off? To emit into a single direction for a single frequency, what would be needed (is it a bulls eye grating)?

Response: We appreciate the good questions raised by the reviewer. The temporal coherence of the thermal emitter is related with the bandwidth, i.e. the Q-factor of the optical resonance. In this work, a high Q resonance is the prerequisite for the thermal emitters to provide an alternative solution to the traditional non-dispersive infrared technique with energy-more-efficient light sources. Regarding the spatial coherence, it is related with the number of spatial/Fourier components that are involved. To achieve spatially coherent thermal emission, it is necessary for the structure to support steep dispersion in the momentum space, with different angles corresponding to different output frequencies. That is, a single frequency only outputs in a single direction. Actually, as we mentioned in the first sentence of the abstract, most of the high-Q resonances in local metasurfaces fulfill this requirement because these optical states are embedded in some steep optical bands, as has been discussed in our original manuscript. The example of bull's eye grating proposed by the reviewer, although represents a good work and has been cited in the revised manuscript now, is however not necessary. If one aspires for a higher spatial coherence, one can even manipulate the FBZ folding to make the thermal emitter operating at a steeper part of the dispersion band. We have added a recent reference paper (Nano Lett. 2024, 24, 764–769) to talk about this possibility by using a period-tripling grating.

However, in many cases in practical applications, especially in optical sensing, one cares more about the narrow bandwidth or in other words higher temporal coherence. For thermal emitters with steep dispersion, due to the rainbow effect,

it becomes necessary to use some spatial filters to select the spectral component of interest, which not only increases the system complexity but also implies a waste of radiation energy. In addition, thermal radiations are usually weak. Large samples are then required with a focusing lens to fully collect all the radiations from the sample. At the presence of the rainbow effect, the superposition of multiple peaks will lead to a broadband response, significantly reducing the temporal coherence.

Our main objective in this work is to address the above problem by harnessing a flat band design with simultaneously high Q-factors over a large range of wavenumbers. This ensures that when a lens with a certain numerical aperture is used, the output emission operating at the same frequency but multiple angles can be collected, without deteriorating the temporal coherence or wasting spectral energy over a wide range of output angles. Based on the requirements of this practical application, we propose a new method to achieve high Q flat band. Through simple structural design and well-developed fabrication technology, we have achieved a thermal emitter with high temporal coherence and emission wavelength independent of the output angle, which will have important advantages in practical applications. To the best of our knowledge, this is the first time reporting this type of thermal emitters, which hold important promises for the thermal emitters to enter the real-world applications.

We have discussed this in the Introduction section and added new References:

“...In a ground-breaking study in 2002, Greffet et al. demonstrated that surface phonon polaritons (SPhPs) supported by the polar material of silicon carbide (SiC) can help achieve coherent thermal emission¹⁵. Later, Inampudi et al. conducted a detailed study on the unidirectional emission performance of thermal emitters based on this platform^{16,17}. However, these emitters can only be designed for a limited operating range (e.g., 10.5–12.5 μm for SiC) referred to as the Reststrahlen band^{18,19}. Another approach involves using the moiré effect supported by a dual-layer twisted SiC grating to achieve narrowband tunable thermal emission²⁰. However, the experimental realization of this complex geometric structure poses significant challenges. The bull's eye grating made from metals, as a standalone structure for realizing a narrowband and directional thermal emission by manipulating the propagation of surface plasmon polaritons (SPPs), has also been studied in recent years²¹. Unfortunately, the relatively large propagation loss of surface waves including both SPhPs and SPPs sets an upper bound for the emission temporal coherence...”

“...So, the broadband and omnidirectional thermal fluctuations in the heated metals may couple into all those modes, reaching thermal emissions with the wavelength highly sensitive to the output angle, i.e. the rainbow effect. If one aspires for a high spatial coherence of the thermal emission, a steep dispersion is desired. In that case, one frequency only corresponds to few spatial/Fourier components, which will work in phase to offer a long spatial coherence length³². One can even manipulate the FBZ folding to make the thermal emitter operating at a steeper part of the dispersion band in order to achieve a higher spatial coherence. However, in practical applications, one cares more about the bandwidth or in other words temporal coherence. For thermal emitters with steep dispersion, due to the rainbow effect, it becomes necessary to use some spatial filters to select the spectral component of interest, which not only increases the system complexity but also implies a waste of radiation energy. In addition, since thermal radiations are usually weak, large samples are preferred to increase the output power, and one normally uses lenses to collect the emitted signals within a large collection angle to fully make use of all the radiations from the whole structure. Due to the rainbow effect, multiple resonances will be collected and the overlap of these spectral components will inevitably lead to a broader resonance, significantly deteriorating the temporal coherence. In this context, a flat band design to host many high-Q optical resonances all operating at the same frequency but a large wide of wavenumbers will help address the problem, and the development of this kind of rainbow-free thermal emitters become necessary.”

[21] Park, J. H., Han, S. E., Nagpal, P. & Norris, D. J. Observation of Thermal Beaming from Tungsten and Molybdenum Bull 's Eyes. *ACS Photonics* **3**, 494–500 (2016).

[32] Sun, K., Levy, U. & Han, Z. Exploiting Zone-Folding Induced Quasi-Bound Modes to Achieve Highly Coherent Thermal Emissions. *Nano Lett.* **24**, 764–769 (2024).

5. The content is here to make this an impactful publication, but the presentation is not really there yet. I think the first figure could use schematics illustrating the "regular" metasurfaces with parabolic dispersion and the flatband structure, including suggested setups of large area vs smaller area + collimating lens. Additionally, I would try and make the figures a little cleaner, which can partially be achieved by not using jet/rainbow color scales.

Response: We thank the reviewer for a valuable input regarding the presentation of our central idea. We have updated Fig.1 and added Fig. S16 to discuss the influence of the metasurface size to the emission performance. The following description has been added to the revised manuscript and Supplementary Information. In addition, we have also removed some colored backgrounds from the data results based on the suggestions of the reviewer to make the data cleaner.

“Fig. 1 (a) briefly illustrates the distinctive spectral responses of high-Q nonlocal metasurfaces that support different dispersion behaviors. For those high-Q modes within steep dispersion bands, the resonance frequency exhibits a strong dependence on the wavevector. When the metasurface of this kind is illuminated by a focused light beam, the excitation of multiple resonances will overlap and give rise to a broadband response in the far-field spectra. For an ideal flat band, however, the modes share almost the same frequency within a wide range of wavevectors, which helps maintain the narrowband characteristics in the far-field spectrum. The slow light effect enabled by this flat dispersion can also enhance the interaction between light and matter, which facilitates the coupling of thermal fluctuations into the nonlocal metasurfaces, thus providing an important foundation for achieving ultra-narrow band and rainbow-free thermal emitters.”

Fig.1 | (a) Schematic diagram illustrates the spectral response of high-Q nonlocal metasurfaces with different dispersion behaviors when excited by a focused beam. (b) Artistic rendering of the thermal emitter, where the red output beam represents the thermal radiation at the QBIC resonance. (c) illustrates the evolution process from the traditional SDS to the final DDS structure, with red and blue block diagrams representing the real parts of E_x and E_y and the dashed line representing the axis of symmetry in the polarization direction. (d) The corresponding shrinking of the FBZ due to the introduction of the perturbation.

“Fig. S16 provides schematically a comparison between the performances of finite and infinite size metasurfaces as thermal emitters. Both finite and infinite structures supporting rainbow bands exhibit broadband emission responses. For finite sized metasurfaces supporting this kind of dispersion band, the truncation of sample is related with more wavenumbers, which leads

to an excitation of multiple resonances. The overlap of them will deteriorate the Q-factor, giving rise to a degradation of the temporal coherence. For flat band emitters, the requirement on the metasurface size is less stringent, considering that the flat-band is always associated with the slow-light effect.”

Fig.S16 | Comparison diagram of emission performance between finite and infinite size metasurface structures.

We have also added the following discussions regarding the size of metasurfaces in the conclusion:

"... Furthermore, the high-Q resonances like QBIC or QGM as the core of nonlocal metasurfaces require a collective interaction between a large number of periods, which suggest the use of a large sample footprint. The slow light propagation associated with the flat-band design possesses less requirement on the size of metasurfaces (see Fig. S16), which makes this rainbow-free thermal emitter more interesting..."

Smaller comments:

- Line 47: I would say that thermal radiation is generally “unpolarized” rather than “polarization independent”.
- Fig. 3D: the $|E|$ and $|H|$ are not really legible on top of the color legend.
- Fig. 4B/D: these seem redundant.
- Fig. 5, line 319: “depdent” should be “dependent”

Response: We greatly appreciate the reviewer’s careful reading and constructive suggestions. We have made proper changes to eliminate the above errors in the revised manuscript. Fig. 4(b) presents the absolute signals although with arbitrary units at different temperatures, which also include the relationship between the emission intensity and the temperature. In contrast, Fig. 4(d) shows the emissivity spectra by normalizing the emission from the sample to that from a standard blackbody of the same size and heated to the same temperature. We hope the reviewer can allow us to keep both of them to provide the reader with full information about the emission performances of our thermal emitter.

Reviewer #3:

The paper is an interesting work combining BIC with for thermal emission control, namely wavelength. Few questions and comments:

Response: We greatly appreciate the positive feedback from the reviewer and for the time on our work. We are pleased to hear that the reviewer finds our paper interesting, especially in its combination of BIC with thermal emission control. We appreciate all the insightful questions and comments from the reviewer, which have been carefully considered in the revised manuscript with necessary revisions to address the concerns. The reviewer's expertise is invaluable to us, and we are grateful for the opportunity to enhance the quality of our manuscript based on the feedback.

1. The authors discuss the need for narrowband emission but it is not very clear the benefits. In other words usually one is interested to control a wideband range of emissions from the source (to save the efficiency)

Response: We greatly appreciate the reviewer for the careful reading and raising valid question. We have added some new discussions in the introduction section, highlighting the importance of narrowband thermal emission.

“...In particular, thermal emission with steered properties has recently garnered significant attentions because they can offer both fundamental interest, e.g. higher temporal and spatial coherence, and energy-more-efficient solutions to many practical applications. For example, traditional non-dispersive infrared (NDIR) system¹⁴ often uses micro light bulbs as light sources to detect the absorption of various gases and compounds. However, many target substances have sharp absorption spectrum, and require the incorporation of narrowband filters in the NDIR system, which not only increases the system complexity, but also greatly reduces energy utilization efficiency since much of the radiations is filtered out and wasted. Therefore, it is crucial to develop narrowband (which suggests high temporal coherence) thermal emitters as energy-more-efficient substitutes with improved spectral selectivity...”

2. The authors discuss other alternatives like to have two beams and use filter to remove one of them, but then one can make proper design with only one beam direction, please see below papers,

S. Inampudi et al, “Unidirectional thermal radiation from SiC metasurface,” JOSA B.

S. Inampudi et al, “Tunable wideband-directive thermal emission from SiC surface using bundled graphene sheets,” Phys. Rev. B.

It is recommended the authors may include then in references due to relevance

Response: We greatly appreciate the relevant papers the reviewer provided, which have been cited in our revised manuscript to strengthen the connection between our results and previous researches. These papers work as a beneficial supplement to enhance the background and relevance of our research.

“...Currently, various micro/nanostructures and physical mechanisms have been employed to achieve narrowband thermal emitters. In a ground-breaking study in 2002, Greffet et al. demonstrated that surface phonon polaritons (SPhPs) supported by the polar material of silicon carbide (SiC) can help achieve coherent thermal emission¹⁵. Later, Inampudi et al. conducted a detailed study on the unidirectional emission performance of thermal emitters based on this platform^{16,17}.”

[16] Inampudi, S., Cheng, J., Salary, M. M. & Mosallaei, H. Unidirectional thermal radiation from a SiC metasurface. J. Opt. Soc. Am. B 35, 39 (2018).

[17] Inampudi, S. & Mosallaei, H. Tunable wideband-directive thermal emission from SiC surface using bundled graphene sheets. Phys. Rev. B 96, 1–7 (2017).

We have also mentioned in our manuscript about the limit of SPhP-based thermal emitters using "Unfortunately, the relatively large propagation loss of surface waves including both SPhPs and SPPs sets an upper bound for the emission temporal coherence." Furthermore, as we responded to the first two reviewers, the simultaneous realization of flat band dispersion and

the high Q-factor is the objective of our work, which can significantly facilitate the utilization of narrow-band thermal emissions with a focusing lens to collect more radiations to increase the output power. These two characteristics may go beyond the capability of SiC-based thermal emissions.

3. Talking about efficiency, the BIC is very high Q, then even small loss can result very low efficiency. Perhaps better the authors show results, without normalization.

Response: We thank the reviewer for the good suggestion. In principle, the BIC has an infinite radiation Q-factor. After being converted to a QBIC, the Q-factor can also be steered to a very high level depending on the magnitude of perturbation. However, for the amorphous materials obtained by electron beam evaporation, the absorption loss cannot be ignored. According to the temporal coupled mode theory (TCMT), perfect emission can only be achieved when the Q_{rad} factor and Q_{abs} factor reach the critical coupling condition. Although we did consider the absorption loss of materials in the calculation, and designed the structural parameters to achieve perfect emission in Fig. 3, the measured normalized emissivity of our fabricated structure can only reach around 50%. We attribute the discrepancy to some additional roughness caused by the imperfect sidewall etching, which further decreases Q_{abs} and leads to a mismatch between Q_{rad} and Q_{abs} . We note that if more advanced material growth techniques like atomic layer deposition or molecular beam epitaxy are used to achieve lower loss dielectric materials, a better agreement between the numerical and experimental results can be obtained.

As suggested by the reviewer, we presented the emission spectra without normalization at different temperatures in Fig. 4(b). A similar demonstration for the thermal emissions from our flat band design and the regular QBIC based structure is presented in Fig. 5. These results demonstrate that our flat band thermal emitter is indeed superior and may have important application prospects in achieving efficient thermal emitters.

4. Isn't the design polarization sensitive? Then for a thermal heat 50% of the power may not be manipulated properly?

Response: Thanks for the insightful question. In fact, the designed thermal emitter in this work can only be capable of output linearly polarized thermal emissions in the y -direction, indicating significant polarization sensitivity in thermal emissions. The thermal fluctuations generated in the underlying metal is usually broadband, omnidirectional, and unpolarized, so only thermal fluctuations with y -directional polarization components can couple with the high-Q resonant mode supported by the upper nonlocal metasurface. The thermal energy of the x -direction polarization component will not be manipulated or utilized. That's a result of the involved high-Q optical resonances, which is essential to achieve thermal emissions with high temporal coherence. Although this indeed makes use of only 50% of the energy, it is still much more efficient compared to regular thermal emitters with rainbow effect where the use of a spatial filter can only retrieve a much smaller proportion of the emitted radiations. Actually, as we responded to the third question from the second reviewer, to achieve thermal emissions with the bandwidth at the order of 10 nm, the scheme of using rainbow-free thermal emitters with a focusing lens can increase the output power by roughly two orders of magnitude, compared to the case of using a regular rainbow thermal emitter with a spatial filter.

5. How sensitive fabrication is to the required design? Given BIC what is tolerance in fabrication within the elements?

Response: We thank the reviewer for the careful reading and bringing up a good question. We used a 50 kV EBL system (Raith Voyager™) in our lab to fabricate the thermal emitter devices. Usually, for a given structure size, the manufacturing error of the machine is within the range of 10 nm. In order to provide a more detailed explanation of the impact of fabrication errors on the emission spectra, we conducted more calculations.

We assumed a fabrication error of 20 nm in the length of the disk, which is fully within the capability of most EBL systems, and calculated the emission spectra, group velocities, and Q values when the long or short axes of the structure increases or decreases by 20 nm. Our calculations show that a spectral shift of approximately 0.04 μm can be observed away from the objective target wavelength of 5.131 μm . However, as one can see from the results in Fig. S2(b) and Fig. 4(b), the temperature change can easily induce such a level of spectral shift. This shows that the spectral shift due to the presence of

fabrication errors can be easily compensated by the change of temperature.

The following content has been added to the Supplementary Information:

“Fig. S15 presents the calculated influence of fabrication errors on the emission spectrum. A schematic diagram of the structure with an increase or decrease of 20 nm in the long/short axis of the elliptical disks is shown in Fig. S15(a). The calculated spectral shifts under these two variations in (a) are presented in Fig. S15(b). For the fabrication error of 20 nm in either dimension, it can be seen that a spectral shift of approximately 0.04 μm can be observed away from the objective target wavelength of 5.131 μm . It is seen from the results in Fig. 4(d) that the center wavelength of the emission spectrum from our experimentally fabricated structure is 5.112 μm at 175 $^{\circ}\text{C}$ and 5.163 μm at 250 $^{\circ}\text{C}$. This shows that the spectral shift due to fabrication errors can be easily compensated by the change of temperature. In addition, through calculations, it can be seen that the dispersion group velocity and Q-factor remain almost unchanged under an error of 20 nm. This indicates that the structure has a high tolerance on the fabrication errors.”

Fig. S15 | (a) A schematic diagram of a structure with an increase or decrease of 20nm on the long/short axis. (b) The emission spectra of structures with different sizes, where the orange line represents the emission spectra of the original design structure size, the red line represents the emission spectra of structures with a reduction of 20nm, and the blue line represents the emission spectra of structures with an increase of 20nm. (c) The calculation results of dispersion group velocities supported by the structures in (a). (d) The dependences of Q-factor on the wavevector for three different structures with 20 nm fabrication errors are shown.

We have added the following discussions in the Conclusion section to talk about the influence of fabrication error to the emission performances:

"...Another thing which may raise concern in practical applications is the spectral shift of central thermal emission wavelength due to fabrication errors. We assume an error of 20 nm, which is fully within the capability of current nanofabrication facilities and conducted some calculations. The results in Fig. S15 demonstrate that with such an error, the spectral shift is only 0.04 μm , which can be easily compensated by the change of temperature according to the results in Fig. 4(b)."

The reviewer can recommend publication after careful addressing the above comments.

Response: We would like to thank you again for your thorough review and constructive comments. We have carefully addressed

all the points raised, and we believe the manuscript is now significantly improved accordingly.

On behalf of all authors,
Zhanghua Han

REVIEWERS' COMMENTS

Reviewer #1 (Remarks to the Author):

The authors (Kaili Sun et al.) of the manuscript "Ultra-narrowband and rainbow-free mid-infrared thermal emitters enabled by a flat band design in distorted photonic lattices" have rigorously modified the manuscript in line with the our suggestions and have responded positively to all the comments. The revised version of the manuscript could be accepted for publication.

Reviewer #2 (Remarks to the Author):

I appreciate the effort the authors have taken into addressing the concerns of all reviewers. I think the work has improved significantly both in content and presentation, and recommend publication.

Reviewer #3 (Remarks to the Author):

The authors have addressed the questions raised. But some issues still remain, like polarization dependence of the design and loosing 50% of the power.

Second Response to Reviewer Comments

Reviewer #1 (Remarks to the Author):

The authors (Kaili Sun et al.) of the manuscript "Ultra-narrowband and rainbow-free mid-infrared thermal emitters enabled by a flat band design in distorted photonic lattices" have rigorously modified the manuscript in line with the our suggestions and have responded positively to all the comments. The revised version of the manuscript could be accepted for publication.

Response: We thank the reviewer for the efforts on our work and the positive recommendation.

Reviewer #2 (Remarks to the Author):

I appreciate the effort the authors have taken into addressing the concerns of all reviewers. I think the work has improved significantly both in content and presentation, and recommend publication.

Response: We thank the reviewer for the efforts on our work and the positive recommendation.

Reviewer #3 (Remarks to the Author):

The authors have addressed the questions raised. But some issues still remain, like polarization dependence of the design and loosing 50% of the power.

Response: We thank the reviewer for the efforts on our work and the positive recommendation.

We acknowledge the concerns raised by the reviewer. However, for many applications requiring linearly polarized light, the use of a polarizer also results in a 50% reduction of the light source's energy. In our work, although the thermal emitters with linearly polarized output characteristics utilizes only 50% of the energy, but compared to traditional thermal emitters with a spatial filter to ensure the temporal coherence (this point has been discussed in our previous response to Reviewer #2 and in section VI of the supplementary information), its efficiency has been significantly enhanced.

On behalf of all authors,

Zhanghua Han